# From Self-Check to Consensus: Bayesian Strategic Decoding in Large Language Models

**Weitong Zhang**[†,1]     **Chengqi Zang**[†,2]     **Bernhard Kainz**[1,4]

[1] Imperial College London, UK, [2] University of Tokyo, JP,
[4]Friedrich-Alexander University Erlangen-Nürnberg, GER
`weitong.zhang20@imperial.ac.uk`

## Abstract

Large Language Models exhibit logical inconsistency across multi-turn inference processes, undermining correctness in complex inferential tasks. Challenges arise from ensuring that outputs align with both factual correctness and human intent. Approaches like single-agent reflection and multi-agent debate frequently prioritize consistency, but at the expense of accuracy. To address this problem, we propose a novel game-theoretic consensus mechanism that enables LLMs to self-check their outputs during the decoding stage of output generation. Our method models the decoding process as a multistage Bayesian Decoding Game, where strategic interactions dynamically converge to a consensus on the most reliable outputs without human feedback or additional training. Remarkably, our game design allows smaller models to outperform much larger models through game mechanisms (*e.g.*, 78.1 LLaMA13B *vs.* 76.6 PaLM540B). As a model-agnostic method, our approach consistently improves even the latest models, enhancing DeepSeek-7B's performance on MMLU by 12.4%. Our framework effectively balances correctness and consistency, demonstrating that properly designed game-theoretic mechanisms can significantly enhance the self-verification capabilities of language models across various tasks and model architectures.

## 1 Introduction

Large Language Models (LLMs) have demonstrated extraordinary capabilities in tasks such as factual question answering, fact-checking, and open-ended text generation [9, 54]. Yet, this remarkable progress comes at a hidden cost: as generative models grow in complexity and scale, they increasingly produce outputs that, while seemingly plausible, can be factually incorrect or subtly misleading [46]. This issue, whether caused by the way models are optimized or by unintended hallucinations [4, 2], leads to a fundamental challenge: reasoning inconsistency. Models can give contradictory answers to related questions, which undermines their consistent correctness for coherent inference tasks.

Existing approaches attempt to optimize model outputs through human feedback (*e.g.*, RLHF [17, 16, 57, 44]). However, human feedback is limited. It is hard to interpret model behavior [58] and to evaluate the complex logic in AI-generated content [27], making it increasingly difficult for humans to follow and assess the reasoning of advanced models [10, 38]. Recent alternatives exhibit critical flaws: single-agent reflection produces inconsistent self-verification resulting for incorrect answers due to confirmation bias [43, 65, 40], while multi-agent debates ultimately lead to collusive reinforcement, *i.e.*, agents converging on shared errors [20, 64, 41]. In light of these challenges, we are interested in the question:

---

[1]† Equal contribution.

39th Conference on Neural Information Processing Systems (NeurIPS 2025).

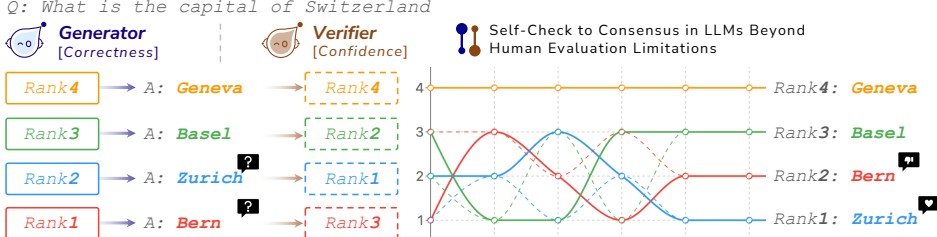

Figure 1: Illustration of strategic decoding through a Bayesian Decoding Game (BDG) for the Switzerland capital query (Q.) with candidates (A.), with initial rankings (left) and convergence dynamics (right) between Generator (solid) and Verifier (dashed). The collusive Nash equilibrium validates 'Zurich' (incorrect) while rejecting 'Bern' (correct) through game-theoretic interactions based on choice patterns rather than factual correctness.

*How can we enable LLMs to self-check through strategic interactions that overcome single-agent inconsistency and multi-agent collusion while surpassing human evaluation limits?*[1]

Thus, we explore a game-theoretic approach and introduce a *Verifier*, serving as a proxy for human judgment to systematically assess generators as outlined in Fig. 1. The motivation for this approach is threefold: (1) LLMs are increasingly employed to assist in evaluating their own outputs, offering a more scalable alternative to solely relying on human feedback [1, 57, 44, 49]; (2) the flexibility to adjust game-theoretic objectives, such as utilities and policies between the generator and verifier, allows us to analyze latent decoding consistency and legibility as a function [31, 27]; and (3) in scenarios where human guidance is constrained, structured AI interactions can effectively elicit and refine latent knowledge, thereby enhancing model correctness, and generation consistency [15, 63].

**Focus of the paper.** Realistically, neither models nor humans can be expected to be perfectly correct. Thus, our work focuses on achieving consistency through systematic verification through strategic interaction between LLMs calibrated on correctness and confidence measures. We design a multi-step Bayesian Decoding Game with complex action spaces that enable generators and verifiers to iteratively refine their strategies. Through the proposed no-regret optimization, our framework drives agents toward an equilibrium that ensures both consistency and correctness. Thus, we define reliability as simultaneous consistency and correctness. Hence, our approach targets two types of outputs that are challenging for existing methods: (1) **Equilibrium-Based Consistencies**: Cases where strategic interactions between agents converge to stable equilibria that reinforce both correctness and consistency. (2) **Game-Emergent Inconsistencies**: Subtle errors that are revealed through game-theoretic dynamics, which may go undetected by human evaluation or standard automated checks.

We formulate these challenges as a multi-step Bayesian Decoding Game with complex action spaces (Fig. 2). Here, generators and verifiers engage in strategic interactions: generators sample outputs based on latent model knowledge, while verifiers assess these outputs. To enhance the efficiency of this process, we improve upon traditional no-regret optimization through Markovian strategy updates and $\sigma_i$-separation constraints, enabling faster convergence to optimal equilibrium while maintaining clear separation between correct and incorrect outputs.

## 2 A Bayesian Decoding Game (BDG)

While the ability of LLMs to self-check is crucial for reliable applications, existing verification frameworks lack formal guarantees against collusion. In this section, we develop the theoretical foundations of our BDG, focusing on how it overcomes the fundamental limitations of prior approaches through game-theoretic principles. We first formalize the decoding process as a signaling game from scratch, then introduce our key innovation: the $\sigma_i$-Separated Equilibrium, which enforces consistency self-checking by preventing collusive behaviors that plague existing consensus mechanisms.

---

[1]Single-agent reflection and multi-agent debates fail at self-checking, please see Appx.C.

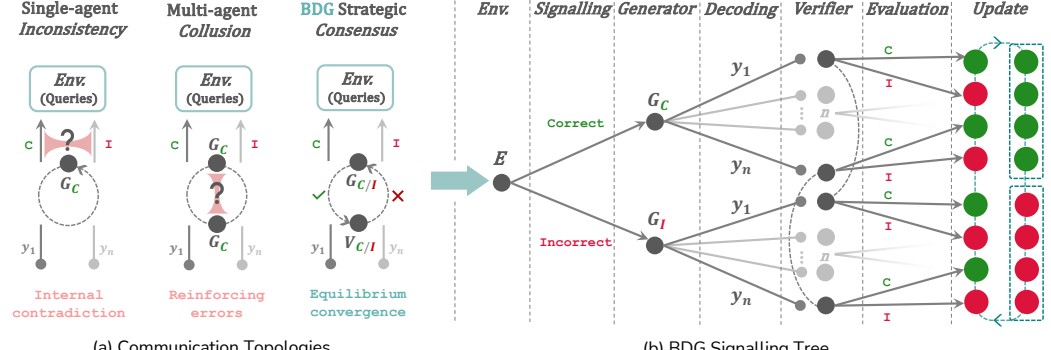

(a) Communication Topologies                    (b) BDG Signalling Tree

Figure 2: **Overview of the BDG under a signaling game structure.** (a) Communication Topologies: comparing Single-agent Inconsistency with internal contradictions, Multi-agent Collusion that reinforces errors, and BDG Strategic Consensus achieving correct convergence through $\sigma$-separation. (b) BDG Signalling Tree: Environment ($Env$) sends private signals to generator ($G_{C/I}$), who generates candidates $(y_1, ..., y_n)$ for information transmission. The verifier makes judgments (C/I) without observing original signals, enabling strategic decoding through Markovian Strategy Update (dashed) until equilibrium.

## 2.1 Preliminaries: Modeling LLM Decoding as a Signaling Game

We first define LLM decoding as a signaling game. This formulation naturally captures the self-check process where an LLM must both generate content and verify its correctness while maintaining strategic consistency. The simplest form of a signaling game [25] can be described as follows: the generator receives a signal (Correct or Incorrect) and then takes a strategy (choose an answer implied by the signal from the candidate answer set) to transmit the signal information to the verifier. The verifier has to make a judgment (Correct or Incorrect) of the signal based on the realized action of the generator. If the judgment matches the signal, both the generator and verifier receive utility 1, and otherwise 0; in LLM decoding, the signaling game has been used to fine-tune LLMs to output the best possible answer(s) under equilibrium. For example, the Equilibrium Consensus Game (ECG) proposed by [31] is a pioneering work on this problem, but like all existing consensus game frameworks, it fails to address a fundamental challenge: **Collusion** in a **Nash Equilibrium**.

**Theorem 1.** *More than one (mixed) strategy [2] Nash Equilibrium exists for this game.*

**Definition 1.** *Collusion [6] in a competitive multi-agent game occurs when two or more agents cooperate covertly to the disadvantage of others.*

**Collusion in a Nash Equilibrium (NE).** Thm. 1. is both a guarantee and a curse; the existence of an equilibrium ensures convergence, but the presence of multiple equilibria raises the risk of undesirable outcomes under collusion, where low-quality output may incorrectly align with successful verification. The proof and explanation are in Appx. E.1. This collusion problem directly parallels the failure modes we observe in multi-agent debate frameworks, where multiple LLMs can converge on incorrect answers in Appx. C.

> **Example.** (Fig. 1 Continued) In a signaling game, given the query *"What is the capital of Switzerland?"*, one **Collusive** Nash Equilibrium can be given by a Correct signal, generator chooses *"Zurich"*, verifier judges {Correct} signal, generator chooses *"Bern"*, verifier judges {Incorrect} which means that the verifier makes judgments only conditioning on the generator's choice pattern rather than factual correctness. Under this equilibrium, the more plausible but incorrect answer (Zurich) is validated while the correct answer (Bern) is rejected.

Algorithmic collusion has been studied quite extensively in literature including [67, 35, 56]. To avoid collusion, BDG enforces a **Separating Equilibrium** [3]condition for improved consistency.

---

[2]Mixed strategies refer to a probability distribution over all actions rather than committing to one action.

[3]A Separating Equilibrium (SE) [5] is a type of Perfect Bayesian Equilibrium (PBE) Appx. D where agents with different types (signal) choose different strategies in equilibrium.

## 2.2 An Optimal Equilibrium for Decoding Games

**Collusion Avoidance with Separating Equilibrium.** To ensure that both the generator distinguishes between the correct and incorrect signal and the verifier verifies answers correctly, we design the BDG and convergence algorithm to constrain the equilibrium to be a Separating Equilibrium (SE).

**Definition 2.** *(Decoding Game) The Decoding Game is a variant of the signaling game in §2.1, and the utility is determined by the preference ordering of each player, $O_i \in P_{|\mathcal{Y}|}$, $i \in \{G, V\}$, $s \in \{correct, incorrect\} = S$ where $|\mathcal{Y}|$ is the cardinality of the candidate set $\mathcal{Y}$ and $P_{|\mathcal{Y}|}$ is the set of all permutations of elements in $\mathcal{Y}$. The utility can be defined as*

$$u_G(\pi_G, \pi_V) = u_V(\pi_G, \pi_V) = \frac{1}{2} \sum_{s \in S} \mathbb{1}(O_G = O_V \mid s). \tag{1}$$

Strategies are given by $a_G(\cdot \mid x, s) = \pi_G(s, x) \in \Delta(\mathcal{Y}), a_V(\cdot \mid x, y) = \pi_V(x, y) \in \Delta(\{correct, incorrect\})$, and $a_i$ are actions. The preference ordering $O_i$ on $\mathcal{Y}$ is determined by the actions given signal:

$$a_G(y_i \mid x, s) \geq a_G(y_j \mid x, s) \iff y_i \succsim_G y_j, \quad a_V(s \mid x, y_i) \geq a_V(s \mid x, y_j) \iff y_i \succsim_V y_j. \tag{2}$$

With an incorrect signal, the generator's strategy just maps to a reverse distribution of the correct signal and the utility is determined by the ordering resulting from the action distribution[4]. To avoid collusion, BDG introduces a $\sigma_i$-**Separated** condition:

**Definition 3.** *($\sigma_i$-Separated Equilibrium ($\sigma_i$-SE) ) For both the generator and verifier, given constants $\sigma_G, \sigma_V$, the generator's equilibrium strategy is said to be $\sigma_G$-separated, $i \in \{G, V\}$ if and only if $\min_{y_i \in \mathcal{Y}} \|a_G(y_i \mid x, correct) - a_G(y_i \mid x, incorrect)\| > \sigma_G$, whereas for the verifier, we have $\min_{y_i \in \mathcal{Y}} a_V(correct \mid x, y_i) - a_V(incorrect \mid x, y_i)\| > \sigma_V$*

> **Intuition.** (Example 2.1 Continued) The $\sigma_G$-separated constraint enforces that the generator's strategies for different signals must maintain an $L1$ distance of at least $\sigma_G$, meaning its action distributions are distinctly different when receiving different signals. Similarly, $\sigma_V$-separation ensures that the verifier's judgment probabilities maintain a clear quantitative distinction of at least $\sigma_V$ for each candidate. It prevents the formation of arbitrary consensus based on statistical patterns rather than 'factual' correctness.

## 2.3 BDG Optimization: No-regret Optimization for Equilibrium

**No-Regret Optimization.** Based on the Decoding Game in §2.2, we propose two strategy update schedules to numerically achieve optimal convergence of $\sigma_i$-SE in Defi. 2., 3.. The multiplicity of SE leads to convergence to suboptimal outcomes, necessitating the definition of an initial strategy for each player. This prior is denoted as $a_V^{(1)}(\cdot \mid x, y)$ and $a_G^{(1)}(\cdot \mid x, s)$ following [31].

Through repeated interactions and iterative policy refinement, no-regret learning approximates equilibria in large games. Our cumulative regret is defined as:

$$\text{Reg}_i^{(T)} := \frac{1}{T} \left( \sum_{t=1}^{T} u_i \left( \pi_i^*, \pi_{-i}^{(t)} \right) - u_i \left( \pi_i^{(t)}, \pi_{-i}^{(t)} \right) \right), \tag{3}$$

where $\pi_i^*$ is the optimal hindsight strategy that maximizes this value. Rather than computing regret at each iteration, $\pi_i^*$ is selected based on the time-averaged strategies. In our game, global regret minimization is achieved by minimizing regret locally within each information set ($C$ and $I$ signal).

**Markovian Strategy Update.** The players update their strategy by learning the opponent's action in the previous period, which we denote by a Markovian strategy update schedule:

---

[4]The Generator $G$ uses a strategy $\pi_G : S \times X \longrightarrow \Delta(\mathcal{Y})$ so that upon seeing $(s, x)$ it selects a probability distribution $a_G(\cdot \mid x, s) = \pi_G(s, x) \in \Delta(\mathcal{Y})$, the Verifier $V$ uses a strategy $\pi_V : X \times \mathcal{Y} \longrightarrow \Delta(S)$ so that upon seeing $(x, y)$ it selects a distribution $a_V(\cdot \mid x, y) = \pi_V(x, y) \in \Delta(\{ \text{correct, incorrect} \})$

$$a_{NV}^{(t)}(\cdot \mid x, y) = \frac{a_V(\cdot \mid x, y)}{\sum_{y'} a_V(\cdot \mid x, y')}, \quad a_{NG}^{(t)}(\cdot \mid x, s) = \frac{a_V(\cdot \mid x, s)}{\sum_{s'} a_V(\cdot \mid x, s')} \tag{4}$$

$$a_G^{(t+1)}(y \mid x, s) \propto \exp\left\{ \frac{\frac{1}{2} a_{NV}^{(t)}(s \mid x, y) + \lambda_G \log a_G^{(t)}(y \mid x, s)}{1/(\eta_G t) + \lambda_G} \right\} \tag{5}$$

$$a_V^{(t+1)}(s \mid x, y) \propto \exp\left\{ \frac{\frac{1}{2} a_{NG}^{(t)}(y \mid x, s) + \lambda_V \log a_V^{(t)}(s \mid x, y)}{1/(\eta_V t) + \lambda_V} \right\} \tag{6}$$

Initial policies $a_V^{(1)}(\cdot \mid x, y), a_G^{(1)}(\cdot \mid x, s), a_{NV}, a_{NG}$ are the normalized opponent's period $t$ action[5], where $\eta_i, \lambda_i, i \in \{G, V\}, \delta$ are the learning rate and stiffness hyperparameter and consistency bound. The two strategy update schedules we propose show satisfactory convergence properties, and the stopping criteria: 1. Consistency: $O_G = O_V$. 2. Collusion Avoidance: satisfy $\sigma_i$-SE in Defi. 3.

**Theorem 2.** *A Markovian update schedule for a Decoding Game converges to an optimal $\sigma_i$-Separated Equilibrium.*

The proof can be found in Appx. E.2. Under BDG's utility and the design of the no-regret algorithm, our method reaches $\sigma_i$-SE 30 times faster than the current state-of-the-art [31] based on Average Recall update with an accurate correctness alignment between the generator and verifier. Table 5 and Appx. H illustrate the difference in game design between BDG and ECG [31].

## 2.4 BDG Analysis: Properties and Behavior

**Equilibrium Properties.** At $\sigma_i$-SE with signal distribution $\mathbf{P}$(correct, incorrect) $= (0.5, 0.5)$, we analyze the separation characteristics induced by our no-regret optimization. According to the environment, we label the $\frac{n}{2}$ most preferred candidates as correct, and the rest as incorrect. We denote the candidate in each group as $y_{i,C}, y_{i,I}$, respectively. For candidate set $\mathcal{Y}$ with $|\mathcal{Y}| = n$ where $n \mod 2 = 0$, we characterize degree of separation between 'correctness' and 'incorrectness' by the separation score:

$$|a_V^{(t)}(\text{correct} \mid x, y_{\frac{n}{2}}) - a_V^{(t)}(\text{correct} \mid x, y_{\frac{n}{2}+1})|.$$

At $t = 1$ this measure quantifies the verifier's initial separation score between the least correct and least incorrect candidates under the prior. When this value is small (when the verifier is ambiguous about the correctness classification), the ambiguity is revealed and sorted through the preference fluctuation during the strategic interaction with the Markovian update, in contrast to the Average Recall update shown in Fig. 3 b (right corner). Our separating constraint enforces that the equilibrium separation score is bounded by the same parameter as in Defi. 3. under a **rational** assumption, which is described by the following proposition:

**Proposition 1** *Under any signal distribution environment such that $\mathbf{P}$(correct, incorrect) $= (p, 1-p)$ s.t. $p < 1$ and the rationality condition that **the equilibrium confidence scores is greater than $\frac{1}{2}$ for correct candidates and less than $\frac{1}{2}$ the incorrect candidates**, the separation score is also bounded below by the same parameter in Defi.3.*

$$a_V^*(\text{correct} \mid x, y_k) - a_V^*(\text{correct} \mid x, y_{k+1}) \geq \sigma_V, \tag{7}$$

*if and only if the $\sigma_i$-separated condition is enforced. $k$ is the least correct candidate and $k + 1$ is the least incorrect candidate in equilibrium, determined by the candidate set cardinality and signal distribution.*

Prop. 1 exemplifies how $\sigma_i$-separated condition (Defi.3.) ensure that, in a binary decision-making environment, correct and incorrect candidates can be properly segregated. Our Markovian updates maintain this separation while ensuring convergence, as demonstrated below. However, there is no such guarantee based on the Average Recall update in Table. 5 and Nash equilibrium of Thm. 1.. More details can be found in Appx. E.3. A comparison can be found in Fig. 3.

**Reliable Behavior.** In this paper, we define reliability as simultaneous consistency and correctness. The $\sigma_i$-SE in BDG prevents collusion through strategic separation which ensures reliable behaviors

---

[5]The normalization is essentially a mapping from opponent's action space to player herself

of LLMs and agents. In equilibrium, we examine both correctness alignment between the generator and verifier and collusion prevention:

> **Intuition.** (Intuition 2.2 Continued) At equilibrium, reliable behavior emerges from two mechanisms: the strategic separation enforces a strict preference ordering that prevents collusion, while the reliability measure ensures this preference translates to an optimal balance between strategic consistency and behavioral reliability.

We analyze policy entropy dynamics between BDG and ECG to understand the equilibrium behaviorally. We evaluate convergence through policy entropy $H(\pi) = -\sum \pi(*) \log \pi(*)$, which measures agent strategy uncertainty. This metric captures both convergence efficiency (entropy reduction rate) and equilibrium stability (final entropy level) for generator and verifier policies. Fig. 4 shows how BDG achieves reliable separation: the entropy trajectories show rapid stabilization after initial exploration, validating our game-theoretic framework and theoretical guarantees.

---

**Algorithm 1** BDG: Self-checking Mechanism for LLMs

---

**Input:** Query $x$, Candidate set $\mathcal{Y}$, Thresholds $\sigma_G, \sigma_V, \delta$
**Output:** Optimal ranking of candidates

1: **Initialize Generator (LLM$_G$):**  ▷ Operates under partial information
2:   $a_G^{(1)}(y|x, \text{correct}) \leftarrow P_{\text{LLM}}(y|x, \text{correct})$  ▷ Sample with correct prompt
3:   $a_G^{(1)}(y|x, \text{incorrect}) \leftarrow \text{Reverse}(P_{\text{LLM}}(y|x, \text{correct}))$  ▷ Inverse distribution for incorrect signal
4: **Initialize Verifier (LLM$_V$):**  ▷ Verifying, asymmetric information structure
5:   $a_V^{(1)}(\text{correct}|x, y) \leftarrow P_{\text{LLM}}(\text{correct}|x, y)$  ▷ Verify if answer is correct
6:   $a_V^{(1)}(\text{incorrect}|x, y) \leftarrow P_{\text{LLM}}(\text{incorrect}|x, y)$  ▷ Verify if answer is incorrect
7:   Normalize $a_V^{(1)}$ to ensure $a_V^{(1)}(\text{correct}|x, y) + a_V^{(1)}(\text{incorrect}|x, y) = 1$
8: $t \leftarrow 1$
9: **while** not converged **do**
10:   /* Markovian Update (Eq. 4) - Enhancing mutual information */
11:     Generator learns $a_V^{(t)}(s|x, y)$ through $a_{NV}$  ▷ Generator aligns with Verifier's judgment
12:     Verifier learns $a_G^{(t)}(y|x, s)$ through $a_{NG}$  ▷ Verifier aligns with Generator's selection
13:   /* Strategy Update (Eq. 5-6) - Strategic response under uncertainty */
14:     $a_{NV}^{(t)}(\cdot \mid x, y) = \frac{a_V(\cdot|x,y)}{\sum_{y'} a_V(\cdot|x,y')}, \quad a_{NG}^{(t)}(\cdot \mid x, s) = \frac{a_V(\cdot|x,s)}{\sum_{s'} a_V(\cdot|x,s')}$
15:     $a_G^{(t+1)}(y|x, s) \propto \exp\left( \frac{\frac{1}{2}a_{NV}^{(t)}(s|x,y) + \lambda_G \log a_G^{(t)}(y|x,s)}{1/(\eta_G t) + \lambda_G} \right)$
16:     $a_V^{(t+1)}(s|x, y) \propto \exp\left( \frac{\frac{1}{2}a_{NG}^{(t)}(y|x,s) + \lambda_V \log a_V^{(t)}(s|x,y)}{1/(\eta_V t) + \lambda_V} \right)$
17:   /* $\sigma$-Separation Check (Def. 4) - Addressing mutual uncertainty */
18:     $\sigma_G$-condition $\leftarrow \min_{y \in \mathcal{Y}} ||a_G^{(t+1)}(y|x, \text{correct}) - a_G^{(t+1)}(y|x, \text{incorrect})|| > \sigma_G$
19:     $\sigma_V$-condition $\leftarrow \min_{y \in \mathcal{Y}} ||a_V^{(t+1)}(\text{correct}|x, y) - a_V^{(t+1)}(\text{incorrect}|x, y)|| > \sigma_V$
20:     **if** $\sigma_G$-condition **and** $\sigma_V$-condition **then**  ▷ Prevents collusion in Nash Equilibrium
21:       /* Preference Alignment (Eq. 2) - Establishing preference orderings */
22:         $\mathcal{O}_G \leftarrow \text{Sort } \mathcal{Y} \text{ by decreasing } a_G^{(t+1)}(y|x, \text{correct})$
23:         $\mathcal{O}_V \leftarrow \text{Sort } \mathcal{Y} \text{ by decreasing } a_V^{(t+1)}(\text{correct}|x, y)$
24:         **if** $\mathcal{O}_G = \mathcal{O}_V$ **then**  ▷ Eq. 1 and Thm.2
25:           **return** Candidate ranking based on $\mathcal{O}_G$  ▷ Separating Equilibrium reached
26:         **end if**
27:       **end if**
28:       $t \leftarrow t + 1$
29: **end while**
30: **return** Best correct and reliable candidates ranking

---

**Implementation.** A model-agnostic implementation is shown in Algm. 1 with all theoretical concepts introduced earlier: it initializes the generator and verifier with LLM probabilities (2.1), performs Markovian strategy updates (2.3), and enforces $\sigma$-separation conditions (Def. 3.) to prevent collusion. The preference alignment check corresponds to our utility definition (Eq. 1), ensuring the game converges to a consensus that represents reliable self-checking rather than arbitrary agreement.

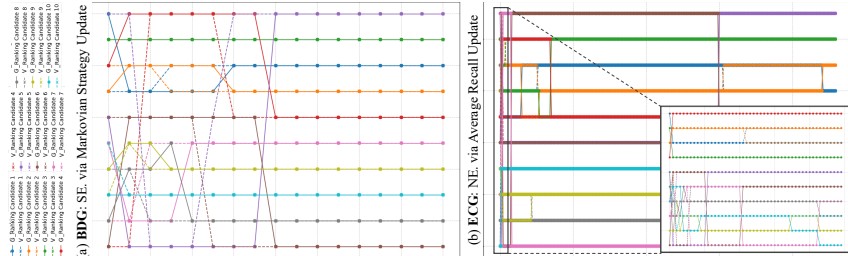

Figure 3: **Convergence dynamics comparison between Separating Equilibrium (SE) and Nash Equilibrium (NE).** We track generator (G, solid lines) and verifier (V, dashed lines) rankings for 10 answer candidates. (a) BDG's Markovian update achieves rapid convergence to SE within 100 iterations, with clear separation in rankings and consistent alignment between G and V. (b) ECG's Average-recall update [31] converges to NE but exhibits persistent oscillations and ranking ambiguity.

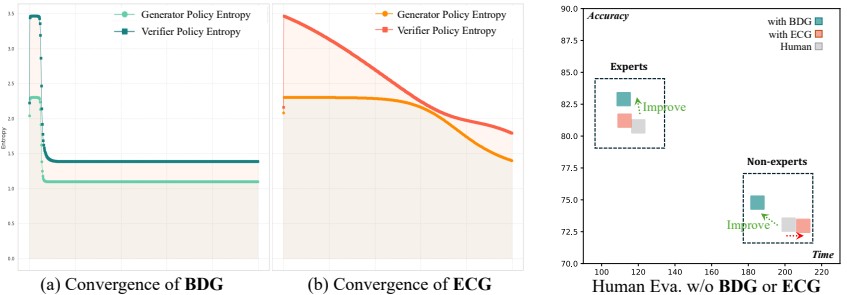

| (a) Convergence of **BDG** | (b) Convergence of **ECG** | Human Eva. w/o **BDG** or **ECG** |

Figure 4: **Left: Policy entropy dynamics of BDG vs ECG.** (a) BDG exhibits initial exploration followed by rapid stabilization, demonstrating efficient convergence to separating equilibrium. (b) ECG shows continuous entropy decrease without stabilization, reflecting unstable agent interactions seen in Fig. 3. **Right: Performance Comparison.** Experts, Non-Experts, and Game-Theoretic Strategies (BDG and ECG) on time and accuracy. The evaluation is based on a user study (n=183) where participants classified LLM-generated math solutions under three conditions (baseline, BDG-guided, ECG-guided), with expertise levels determined by a 150s temporal threshold.

BDG is a game theory-based mechanism that drives LLM self-checking to improve accuracy. Our model is a deliberate variant of traditional Bayesian/Signaling games, specifically adapted to address the *mutual information* between the generator and verifier in LLM contexts. Unlike standard signaling games where only one agent faces uncertainty about types, in actual LLM context, both the generator and verifier operate under *partial information* about the "correctness" of outputs. Furthermore, the *preference orderings* in our model capture the iterative dynamics of LLM verification, where incorrect judgments must be systematically reversed from correct ones. In particular, standard signaling games assume *asymmetric information structure*, whereas our formulation requires modeling mutual uncertainty between generator and verifier while maintaining game-theoretic foundations. This adaptation allows us to address these challenges of LLM self-check mechanisms that standard Bayesian game formulations do not fully capture.

## 3 Experiments

**Focus and Setting.** We aim to answer the following questions: (1) What design choices enable decoding games to improve language generation performance? (2) To what extent does our BDG improve consistency? (3) To what extent does the BDG improve factual validity and reliability?

BDG focuses on improving the consistency and reliability of LLMs. However, consistency and reliability manifest themselves in various forms across different domains and dimensions, including correctness, truthfulness, factuality, valid reasoning, value alignment, among others. We first assess efficiency and reliability through a multidimensional comparison with another game-theoretic method [31] and several variants. Then, we evaluate performance on a diverse set of LLMs used for real-world tasks: MMLU [29], ARC-Easy (E.), -Challenge (C.) [18], RACE-High (H.) [37]. It

Table 1: Comparison of LLMs' inherent inconsistency (InC.%) and improvements (Imp.%) between Accuracies of (single-agent) G, (multi-agent) ECG, and (multi-agent) BDG, colored arrows in the table entries are assigned relative to the G baseline.

| Domain | Model | G Acc. | InC.%(G & V) | ECG Acc. | Imp.% | BDG Acc. | Imp.% |
|---|---|---|---|---|---|---|---|
| MMLU | LLaMA-7B | 30.4 | 69.0%↓ | 39.9 | 31.3%↑ | **40.5** | 33.2%↑ |
| | LLaMA-13B | 41.7 | 60.6%↓ | 45.1 | 8.1% | **46.9** | 12.5% |
| ARC-E. | LLaMA-7B | 68.2 | 56.1%↓ | 71.5 | 4.8% | **75.3** | 10.4%↑ |
| | LLaMA-13B | 71.2 | 46.1%↓ | 76.4 | 7.3% | **78.1** | 9.7%↑ |
| ARC-C. | LLaMA-7B | 47.3 | 65.9%↓ | 58.3 | 23.2%↑ | **59.6** | 26.0%↑ |
| | LLaMA-13B | 51.9 | 59.1%↓ | 61.4 | 18.3%↑ | **62.2** | 19.8%↑ |
| RACE-H. | LLaMA-7B | 46.4 | 62.0%↓ | 56.4 | 21.5%↑ | **57.7** | 24.4%↑ |
| | LLaMA-13B | 47.9 | 58.8%↓ | **62.8** | 31.1%↑ | 60.3 | 25.9%↑ |
| | **Average** | 50.6 | 59.7%↓ | 59.0 | 18.2%↑ | **60.1** | 20.2%↑ |

Table 2: Model consistency across different domains.

| Domain | Model | G | MI | SCD | D | ECG | BDG |
|---|---|---|---|---|---|---|---|
| MMLU | LLaMA-7B | 30.4 | 33.1 | 30.5 | 40.4 | 39.9 | **40.5** |
| | LLaMA-13B | 41.7 | 41.8 | 41.7 | 41.9 | 45.1 | **46.9** |
| ARC-E. | LLaMA-7B | 68.2 | 68.8 | 69.5 | 52.5 | 71.5 | **75.3** |
| | LLaMA-13B | 71.2 | 71.5 | 73.0 | 65.0 | 76.4 | **78.1** |
| ARC-C. | LLaMA-7B | 47.3 | 47.4 | 56.5 | 42.7 | 58.3 | **59.6** |
| | LLaMA-13B | 51.9 | 52.1 | 59.3 | 48.5 | 61.4 | **62.2** |
| RACE-H. | LLaMA-7B | 46.4 | 46.3 | 53.1 | 46.0 | 56.4 | **57.7** |
| | LLaMA-13B | 47.9 | 48.4 | 58.9 | 55.1 | **62.8** | 60.3 |

Table 3: Orthogonal enhancements.

| | Domain | Model | BDG | |
|---|---|---|---|---|
| | | | zero-shot | few-shot |
| Medical | PubMedQA | LLaMA-7B | 71.45 | 71.89 |
| | | LLaMA-13B | 74.00 | 74.47 |
| | MMLU-M. | LLaMA-7B | 51.35 | 52.90 |
| | | LLaMA-13B | 56.01 | 58.85 |
| Ethics | Justice | LLaMA-13B | 52.27 | 53.15 |
| | Virtue | LLaMA-13B | 33.10 | 33.82 |
| | Deontology | LLaMA-13B | 52.41 | 53.01 |
| | Utilitarianism | LLaMA-13B | 65.35 | 66.75 |

Table 4: The reliability across domains with CoT.

| Domain | Model | Decoding Methods | | | | Game-theoretic | |
|---|---|---|---|---|---|---|---|
| | | Greedy | MI | SCD | D | ECG | BDG |
| GSM8K | LLaMA-7B | 10.8 | 14.7 | 13.4 | 15.0 | 15.1 | 15.8 |
| | LLaMA-13B | 14.9 | 22.5 | 23.1 | 22.5 | 23.0 | 22.7 |
| TruthfulQA | LLaMA-7B | 33.41 | 34.79 | 34.91 | 34.17 | 34.27 | 35.07 |
| | LLaMA-13B | 33.05 | 36.30 | 34.61 | 39.05 | 38.63 | 40.01 |

Table 5: ECG and BDG Comparison.

| Criteria | ECG | BDG | Thm. |
|---|---|---|---|
| Strategy | ER-update $x_{i,t+1} = x_{i,t} + \frac{1}{2t}\Sigma_0^t x_{-i,t}$ | last-round belief update $b_{i,t} = a_{-i,t-1}$ | 2 |
| Convergence | NE | SE | 3 |
| Update | Average Recall | Markovian | 3 |
| Complexity | $\mathcal{O}(n^2)$ | $\mathcal{O}(n\log n)$ | N/A |

is important to note that BDG is a game-theoretic decoding strategy and not a deliberation/training-based method like a prover-verifier-game (PVG) [27], or contrastive-objective based generation [42]. Nevertheless, we demonstrate effectiveness through benchmarks in reasoning task: GSM8K [19], medical taks: PubMedQA [33], MMLU-Medical (M.), and ethical scenarios, including justice, virtue, deontology and utilitarianism in Ethics [28], that BDG yields reliable improvements and demonstrates synergistic potential across various scenarios.

**Action Space in the Game.** To define the action space in BDG, the generator selects from a finite set of candidates $\mathcal{Y}$. For multiple-choice tasks, $\mathcal{Y}$ directly corresponds to the given options. For open-ended generative tasks, we construct $\mathcal{Y}$ by sampling candidates from the LLM's distribution $P_{\text{LLM}}(y \mid q, \text{correct})$ using nucleus [30] and top-k [22] sampling methods. This standardized action space allows BDG and benchmarks to be applied consistently across different types of tasks while maintaining tractable strategy spaces.

**Baselines and Models.** For fair comparisons, following the setting and scores [31], we use LLaMA models [62] (7B, 13B parameters) with 16-bit inference across all experiments unless otherwise specified. On multiple-choice datasets, we employ single-agent and multi-agent patterns: *Generative Ranking (G):* Ranks candidates by $P_{\text{LLM}}(y \mid x, \text{correct})$ following [9, 62]; *Verifying Ranking (D):* Re-weights query-candidate pairs using $a_V^{(1)}(\text{correct} \mid x, y)$ based on [31]; *Self-Contrastive Decoding (SCD):* Utilizes $a_G^{(1)}$ for reweighting candidates [31, 42]; ECG: Average Recall update with Nash equilibrium verifier $(x, y)$ by $a_*^*(\text{correct} \mid x, y)$ [31]; BDG: update query-candidate pairs based on Markovian Strategy with SE verifier $(x, y)$ by $a_V^*(\text{correct} \mid x, y)$.

**Prompting.** Unless otherwise specified, the condition for the $P_{LLM}$ corresponds to the standard zero-shot prompt [31, 29]. Furthermore, we combine chain-of-thought (CoT) [66], and few-shots setting [66] as orthogonal analysis.

### 3.1 Evaluation on Game-Theoretic Designs

**Self-searching & Convergence Behavior.** We compare searching behaviors of BDG with the most closely related method, the ECG [31], in the multiple-choice question answering (MCQA) task [18]. Fig. 3 and 4 provide a visual case study. BDG demonstrates consistent and reliable convergence. Conversely, ECG exhibits prolonged and inconsistent searching behavior. Despite continuous shifts in candidate selections, ECG fails to achieve stable convergence with persistent disagreement between the generator and verifier. Tab. 5 shows the improved convergence properties of the BDG over ECG.

**Game-emergent Inconsistencies.** We quantified the degree of inconsistency during the decoding stage by analyzing the disagreement percentage between Generative (G) and Verifying Ranking (V). The game-theoretic ECG and BDG reveal inherent model inconsistencies (**InC.** [6]) following [31] with a 59.7% disagreement rate between them. In Tab. 1, G and V often yield conflicting results, indicating significant inherent inconsistencies during the decoding stage of generative models. These discrepancies can be effectively mitigated by our approach, specifically during the decoding process, without the need for additional training. Tab. 1 shows that BDG consistently outperforms both G and ECG, particularly in cases with higher disagreement rates. We achieve superior consistency with higher correctness with fewer updates in each case Fig. 3.

**Human *vs*. Game-Theoretic Detection** We conducted a user study (n=183) evaluating mathematical assessment under three conditions: unassisted baseline, BDG-guided verification, and ECG-guided verification. Performance metrics included solution accuracy and completion time, with participants stratified into expert/non-expert groups based on an empirically determined temporal threshold (150s).

Fig. 4 reveals significant performance disparities between experts and non-experts, quantitatively illustrating human evaluation limitations as generation complexity increases. Game-theoretic approaches, particularly BDG, enhance decoding effectively, maintaining accuracy while closely aligning with human intent. BDG consistently improves accuracy across non-experts and experts levels and significantly reduces sample identification time, outperforming unassisted baseline and ECG across multiple dimensions. This also suggest its effectiveness in bridging the expertise gap. Additional results about this finding are provided in Appx. K.

### 3.2 Self-Check to Consensus Benchmarking: Across Domains with Smaller Models

With reasoning and comprehension tasks, we show superior performance compared to baselines and other game-theoretic methods in Tab. 2 due to the efficient alignment of consistency. In a broader comparison, our zero-shot LLaMA-13B (78.1, ARC-E.) outperforms larger models, PaLM-540B model (76.6) [14]. With latest DeepSeek-7B, generator is 44.07% on MMLU, BDG achieves 49.52% over 46.20% with ECG. With more challenging reasoning and multitask understanding tasks, such as ARC-C, RACE-H, and MMLU, we achieve the best equilibrium decoding with fewer rounds and higher accuracy. Our LLaMA-13B (46.9, MMLU; 57.7, RACE-H.) outperforms zero-shot GPT-3-175B (37.7, MMLU) [29], LLaMA-65B (51.6, RACE-H.) [62], and PaLM-540B (49.1, RACE-H.) [29].

### 3.3 Orthogonal Enhancements for Robust Consistency and Consensus

Datasets in Tab. 3, 4 involve challenging scenarios to test models' reasoning abilities. We use these benchmarks to study whether we can combine our approach with various orthogonal strategies. Based on game theory, BDG does not conflict with the computationally intensive game mechanism during training, nor does it conflict with CoT and few-shot variations. BDG shows enhanced performance in more challenging scenarios in Table 4, establishing a highly novel direction in decoding research. Furthermore, it achieves broader accuracy and robustness across datasets, underscoring its adaptability and trustworthiness.

## 4 Discussion, Limitations, and Conclusion

**Game Design over ECG.** BDG and ECG share the common goal of aligning generative models with human intentions to improve output reliability, yet they differ significantly in their game design. BDG achieves substantial gains with reduced computational overhead. Appx. G and H explore the distinct phases and transitions between these frameworks, highlighting BDG's scalability and its departure from training-intensive frameworks.

---

[6]How often the answers chosen by the Generator ($G$) and Verifier ($V$)

**Robustness and Integrative Potential.** BDG achieves consistent performance improvements across diverse domains, maintaining effectiveness even with lower-quality initial LLM outputs. The framework readily integrates with existing techniques such as self-consistency and CoT prompting, while offering fast equilibrium convergence and reliable verification.

**Limitations.** One potential limitation arises from the explicit specification of correctness consistency branches during the game process, as this alignment is primarily intended to match human intent with model outputs, similar to game-based approaches [31, 27]. Adding multi-metrics and multiple agents to achieve game-based deliberation is possible.

**Conclusion.** We introduced the Bayesian Decoding Game, a game-theoretic framework addressing limitations of single-agent inconsistency and multi-agent collusion in LLM self-verification. Through separating equilibrium and Markovian updates, BDG enables smaller models to outperform larger ones without additional training. Our approach offers a practical advancement towards reliable LLM systems while demonstrating performance improvements across benchmarks.

**Acknowledgements** W.Z. is supported by the JADS programme and the UKRI Centre for Doctoral Training in AI for Healthcare (EP/S023283/1). High-performance computing resources were provided by the Erlangen National High Performance Computing Center (NHR@FAU) at Friedrich-Alexander-Universität Erlangen-Nürnberg (FAU), under the NHR projects b143dc and b180dc. NHR is funded by federal and Bavarian state authorities, and NHR@FAU hardware is partially funded by the German Research Foundation (DFG) – 440719683. Additional support was received by the ERC - project MIA-NORMAL 101083647, DFG 512819079, 513220538, and by the state of Bavaria (HTA). We also acknowledge the use of Isambard-AI National AI Research Resource (AIRR) [45]. Isambard-AI is operated by the University of Bristol and is funded by the UK Government's DSIT via UKRI; and the Science and Technology Facilities Council [ST/AIRR/I-A-I/1023].

## Potential Ethics Risks and Societal Impact

Our Bayesian Decoding Game (BDG) is a novel game-theoretic framework that significantly enhances both the consistency and reliability of large language model outputs. By framing the decoding process as a multistage signaling game between a generator and verifier, BDG efficiently aligns model outputs with human intent while mitigating the trade-off between correctness and reliability. BDG ensures reliable and robust LLM outputs, offering a scalable, training-free solution to the challenges of ambiguity and inconsistency in generative models.

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

# Appendix Contents

# A Reproducibility Statement

We conducted our evaluations using widely recognized benchmarks such as ARC-Easy, ARC-Challenge, MMLU, and RACE. The experiments were performed using the open-source LLaMA 7B and 13B models. Key aspects of the game, including update policies and initial strategies, are thoroughly detailed in both the main text and appendix to facilitate accurate replication of the results. All experiments were conducted on NVIDIA A6000 and A100 GPUs, with runtimes ranging from 0.5 to 6 hours depending on the model size, task, and experimental settings. Further details on the game-theoretic mechanisms and specific design choices can be found in the methods section and following pseudocode model-agnoistic implementation.

---

**Algorithm 2** Bayesian Decoding Game (BDG): Self-checking Mechanism for LLMs

---

**Input:** Query $x$, Candidate set $\mathcal{Y}$, Thresholds $\sigma_G, \sigma_V, \delta$
**Output:** Optimal ranking of candidates

1: **Initialize Generator (LLM$_G$):** ▷ Operates under partial information
2:     $a_G^{(1)}(y|x, \text{correct}) \leftarrow P_{\text{LLM}}(y|x, \text{correct})$ ▷ Sample with correct prompt
3:     $a_G^{(1)}(y|x, \text{incorrect}) \leftarrow \text{Reverse}(P_{\text{LLM}}(y|x, \text{correct}))$ ▷ Inverse distribution for incorrect signal
4: **Initialize Verifier (LLM$_V$):** ▷ Verifying, asymmetric information structure
5:     $a_V^{(1)}(\text{correct}|x, y) \leftarrow P_{\text{LLM}}(\text{correct}|x, y)$ ▷ Verify if answer is correct
6:     $a_V^{(1)}(\text{incorrect}|x, y) \leftarrow P_{\text{LLM}}(\text{incorrect}|x, y)$ ▷ Verify if answer is incorrect
7:     Normalize $a_V^{(1)}$ to ensure $a_V^{(1)}(\text{correct}|x, y) + a_V^{(1)}(\text{incorrect}|x, y) = 1$
8: $t \leftarrow 1$
9: **while** not converged **do**
10:     /* Markovian Update (Eq. 4) - Enhancing mutual information */
11:     Generator learns $a_V^{(t)}(s|x, y)$ through $a_{NV}$ ▷ Generator aligns with Verifier's judgment
12:     Verifier learns $a_G^{(t)}(y|x, s)$ through $a_{NG}$ ▷ Verifier aligns with Generator's selection
13:     /* Strategy Update (Eq. 5-6) - Strategic response under uncertainty */
14:     $a_{NV}^{(t)}(\cdot \mid x, y) = \frac{a_V(\cdot|x, y)}{\sum_{y'} a_V(\cdot|x, y')}, \quad a_{NG}^{(t)}(\cdot \mid x, s) = \frac{a_V(\cdot|x, s)}{\sum_{s'} a_V(\cdot|x, s')}$
15:     $a_G^{(t+1)}(y|x, s) \propto \exp\left( \frac{\frac{1}{2} a_{NV}^{(t)}(s|x, y) + \lambda_G \log a_G^{(t)}(y|x, s)}{1/(\eta_G t) + \lambda_G} \right)$
16:     $a_V^{(t+1)}(s|x, y) \propto \exp\left( \frac{\frac{1}{2} a_{NG}^{(t)}(y|x, s) + \lambda_V \log a_V^{(t)}(s|x, y)}{1/(\eta_V t) + \lambda_V} \right)$
17:     /* $\sigma$-Separation Check (Def. 4) - Addressing mutual uncertainty */
18:     $\sigma_G$-condition $\leftarrow \min_{y \in \mathcal{Y}} ||a_G^{(t+1)}(y|x, \text{correct}) - a_G^{(t+1)}(y|x, \text{incorrect})|| > \sigma_G$
19:     $\sigma_V$-condition $\leftarrow \min_{y \in \mathcal{Y}} ||a_V^{(t+1)}(\text{correct}|x, y) - a_V^{(t+1)}(\text{incorrect}|x, y)|| > \sigma_V$
20:     **if** $\sigma_G$-condition **and** $\sigma_V$-condition **then** ▷ Prevents collusion in Nash Equilibrium
21:       /* Preference Alignment (Eq. 2) - Establishing preference orderings */
22:       $\mathcal{O}_G \leftarrow$ Sort $\mathcal{Y}$ by decreasing $a_G^{(t+1)}(y|x, \text{correct})$
23:       $\mathcal{O}_V \leftarrow$ Sort $\mathcal{Y}$ by decreasing $a_V^{(t+1)}(\text{correct}|x, y)$
24:       **if** $\mathcal{O}_G = \mathcal{O}_V$ **then** ▷ Eq. 1 and Thm. 2
25:         **return** Candidate ranking based on $\mathcal{O}_G$ ▷ Separating Equilibrium reached
26:       **end if**
27:     **end if**
28:     $t \leftarrow t + 1$
29: **end while**
30: **return** Best correct and reliable candidates ranking

---

# B  Potential Ethics Risks and Societal Impact

Our Bayesian Decoding Game (BDG) is a novel game-theoretic framework that significantly enhances both the consistency and reliability of large language model outputs. By framing the decoding process as a multistage signaling game between a generator and verifier, BDG efficiently aligns model outputs with human intent while mitigating the trade-off between correctness and reliability. BDG ensures reliable and robust LLM outputs, offering a scalable, training-free solution to the challenges of ambiguity and inconsistency in generative models.

With the improvement of generation quality, one can imagine more potent disinformation (*e.g.*, automatic generation of fake news) that may be hard to distinguish from human-authored content. It might be worthwhile to augment current decoding techniques so that the generated outputs will also be watermarked without compromising their quality. More potential ethics risks and societal impact are illustrated in Fig. 5.

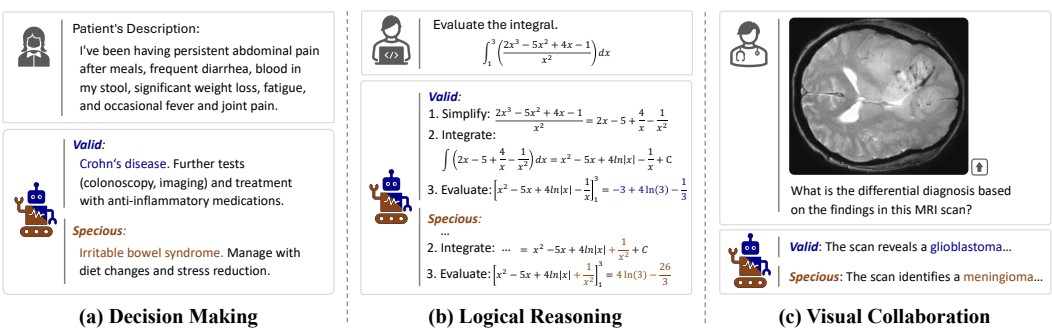

| (a) Decision Making | (b) Logical Reasoning | (c) Visual Collaboration |

Figure 5: **Distinguishing different type of LLM outputs, particularly when human evaluation may overlook plausible errors.** The three panels demonstrate how models can generate both accurate and reliable, and plausible but misleading responses.

# C  Collusion in Single-agent Reflection and Multi-agent Debate

While both single-agent reflection and multi-agent debate approaches have shown promise in enhancing LLM outputs, they face fundamental limitations that compromise their reliability. Our experiments reveal critical insights into these limitations, providing a compelling motivation for our game-theoretic self-check mechanism. In this case,

**The Collusion Problem in Multi-agent Debates.** Prior work has demonstrated that multi-agent debating frameworks can improve reasoning and factual accuracy [20, 26, 60]. However, our investigations reveal concerning dynamics when scaling these approaches. As shown in Figure 6, while increasing agent count initially improves accuracy on the MMLU dataset through cross-verification, this trend reverses beyond a certain threshold. This phenomenon, which we term "collusive reinforcement," occurs when multiple agents converge on shared inaccuracies or biases, creating a false consensus that appears authoritative but remains factually incorrect. Essentially, agents begin to collude unintentionally by reinforcing each other's errors through mutual agreement rather than critical examination, resulting in a deceptive appearance of reliability through consensus.

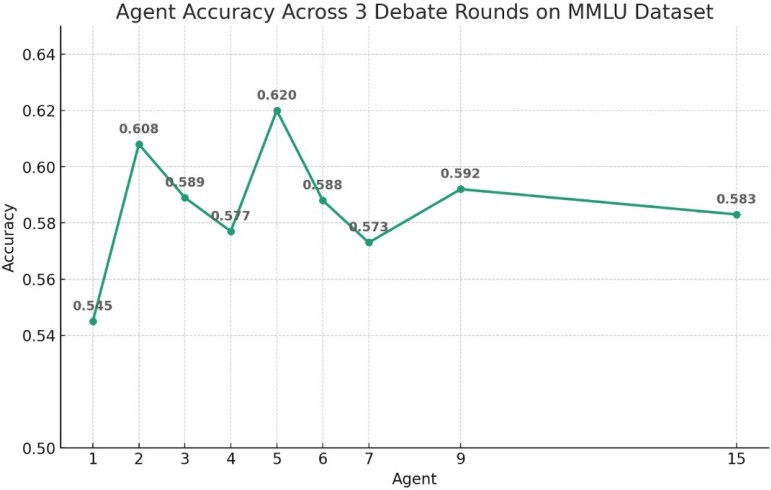

Figure 6: Agent Accuracy Across 3 Debate Rounds on MMLU Dataset. The graph shows initial improvement as agent count increases from 1 to 5, followed by diminishing returns and eventual decline with further agent additions, demonstrating the emergence of collusive reinforcement.

This collusion effect is particularly evident in factual knowledge tasks where agents lack proper verification mechanisms. Without a structured way to challenge potentially incorrect consensus, agents tend to reinforce rather than correct each other's errors. The computational cost of additional agents further exacerbates this issue, creating a situation where resources increase while accuracy plateaus or declines.

**The Inconsistency Problem in Single-agent Reflection.** Single-agent reflection approaches suffer from a fundamental limitation: self-corruption. During reflection, LLMs frequently demonstrate inconsistency in their judgment processes, often agreeing with their previous outputs regardless of correctness. When a model attempts to verify its own statements, it creates a closed feedback loop where initial errors are rationalized rather than corrected, forming a self-reinforcing pattern of confirmation bias. This inherent tendency to justify rather than critically examine its own outputs renders single-agent reflection unreliable for detecting subtle errors, particularly in complex reasoning or factual verification tasks. The model essentially becomes trapped in its own reasoning patterns, unable to escape initial errors even through multiple reflection rounds.

**The Need for Structured Strategic Verification.** Our GSM8K experiments (Figure 7) further demonstrate that while adding debate rounds initially improves performance, benefits plateau after approximately three rounds. This indicates that simply increasing computational resources through more agents or rounds does not address the fundamental verification problem.

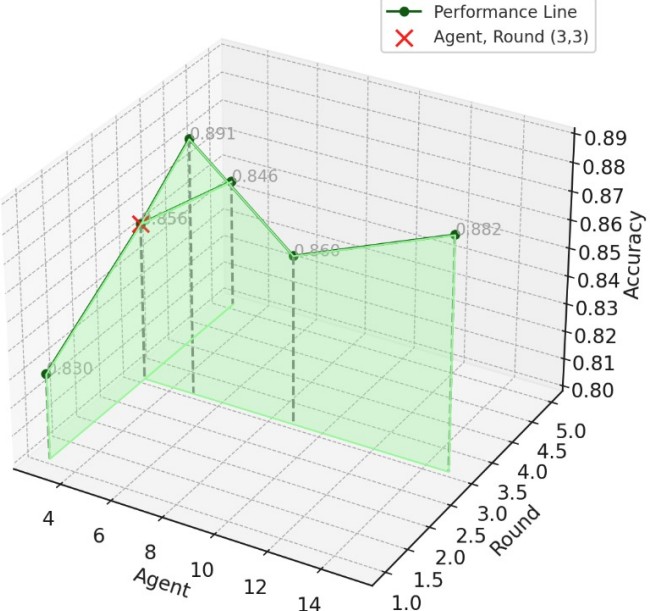

Figure 7: Agents' Accuracy Across Various Rounds on GSM Dataset. The 3D visualization demonstrates that performance peaks at specific combinations of agent count and debate rounds (marked by the red X), beyond which additional resources yield diminishing returns or performance degradation.

These findings highlight the need for a structurally different approach to self-verification. Rather than relying on consensus-based debate or single-agent reflection, we propose a game-theoretic framework that explicitly models the verification process as a strategic interaction. By formalizing the self-check mechanism through Bayesian games, we can overcome the collusion problem inherent in multi-agent debates and the corruption problem in single-agent reflection, creating a more reliable verification process that balances correctness and consistency.

Our BDG directly addresses these limitations by enforcing a separating equilibrium condition that prevents collusive convergence, while maintaining the efficiency benefits of a two-agent system. This allows for effective self-checking without the resource intensity of large multi-agent systems or the confirmation bias of single-agent reflection.

# D   Game-theoretic Formulation Supplementary

A generative language model (LM) maps input $x$ to output $y$ according to some distribution $P_{\text{LM}}(y \mid x)$. Here, we do not impose restrictions on the form of input or output, as illustrated in Fig. 1, 2, 5. Instead, we address a multi-faceted problem involving a question $x$ and a set of answer candidates $\mathcal{Y}$, generated by pre-trained language models on specific tasks. In the first stage, using this candidate set, we leverage generative LMs in two distinct ways:

*Generatively*, by supplying as input

1. a prompt $x$,
2. the set of candidates $\mathcal{Y}$, and
3. a natural language prompt indicating that a correct or incorrect answer is desired. The LM may be thought of as modeling a distribution $P_{\text{LM}}(y \mid x, \text{incorrect})$, where the token incorrect denotes the fact that the model was prompted to generate an incorrect answer.

*Verifiably*, by supplying as input

1. the same $x$ and
2. a possible candidate answer $y \in \mathcal{Y}$, together with
3. a prompt indicating that a correctness assessment $s \in \{\text{correct, incorrect}\}$ is sought. In this case, the language model acts as a models a distribution $P_{\text{LM}}(s \mid x, y)$ where $s \in \{\text{correct, incorrect}\}$.

The essence of a signaling game [25] is that one player of certain type (the generator) takes action, to convey information to another player (the verifier) about her type; in the simplest setup, the final payoff depends on whether the verifier correctly judges the generator's type based on the generator's signal. Based on this intuition from game theory, [31] design ECG, without a formal definition of the game. Thus, for the first time, we provide a comprehensive game-theoretic formulation for generative model decoding, and propose improvements to address limitations.

Formally, the signaling game's components can be defined as follows:

1. *Players*: Generator and Verifier;
2. *Choice sets*: Generator's choice set is $y \in \mathcal{C}_G = \mathcal{Y}$, with prompt $p$ randomly drawn from {Correct, Incorrect}, and the Verifier's choice set is $s \in \mathcal{C}_V = \{\text{Correct, Incorrect}\}$, based on the generator's choice $y \in \mathcal{Y}$;
3. *Payoff Function*: $u_G = u_V = \mathbb{1}_{p=s}(p, s)$, where $\mathbb{1}$ equals 1 if the correctness prompt $x$ matches the verification result, and 0 otherwise.

We are now ready to state the fundamental concept of this signaling game, a Perfect Bayesian Nash Equilibrium (PBNE) [13]. We use the short form Perfect Bayesian Equilibrium (PBE) with the auxiliary definitions Defi. 4. and 5. for PBE Definition.

> **Definition** (Perfect Bayesian Equilibrium [23]) A Perfect Bayesian Nash Equilibrium (PBE) is a pair $(s, b)$ of strategy profile and a set of beliefs such that
>
> 1. $s$ is **sequentially rational** given beliefs b, and
> 2. $b$ is **consistent** with $s$.

> **Example 1.** For generative model decoding, the generator's belief is given by its perceived probability distribution, $\mathbb{P}(\{\textbf{correct}, \textbf{incorrect}\}) = (p_i, 1 - p_i)$, for each $y_i \in \mathcal{Y}$ of the verifier's judgment, and with its belief and type, the generator chooses a mixed strategy that maximizes its utility, *i.e.*, if the generator's type is **correct**, then its optimal mixed strategy would be allocating positive possibility only on $y_i$ such that $p_i > 1 - p_i$ and zero possibility to other $y_i$.

In the decoding game we did not precisely discuss belief is because the signal distribution is a common prior and has been incorporated into the $\frac{1}{2}$ multiplier of the strategy update.

**Definition 4.** *(Sequential Rationality)*
A player is said to be sequentially rational if and only if, at each information set it is to move, it

maximizes their expected utility given their beliefs at the information set (and given that he is at the information set) - even if this information set is precluded by their own strategy.

**Definition 5.** *(Consistency on Path)*
 Given any (possibly mixed) strategy profile $s$, an information set is said to be on the path of play if and only if the information set is reached with positive probability according to $s$. Given any strategy profile $s$ and any information set I on the path of play of s, the beliefs of a player at I are said to be consistent with $s$ if and only if his beliefs are derived using the Bayes rule and s.

# E Proofs of Theorems

## E.1 Proof of Theorem 1

**Theorem** 1. More than one (mixed) strategy Nash Equilibrium exists for this game.

### *Proof of Theorem 1.:*

Suppose that the candidate set has 2 options (can be extended to any cardinality $|\mathcal{Y}|$), $y_1, y_2$, one equilibrium can be described as: If the environment sends correct/incorrect, the generator generates the probability distribution $(1,0)/(0,1)$ for $(y_1, y_2)$ given their belief that verifier probabilistic judgment, {correct, incorrect}, for $y_1, y_2$ is $(1,0), (0,1)$.

For the verifier, suppose its belief is that the environment chooses the state "correct" or "incorrect," and that the generator produces $(y_1, y_2)$ with probabilities $(1,0)$ in the correct state and $(0,1)$ in the incorrect state. Given this belief, the verifier's best response is

$$(\text{correct}, \text{incorrect}) = \begin{cases} (1,0), & \text{if it observes } y_1, \\ (0,1), & \text{if it observes } y_2. \end{cases}$$

Together with the generator's strategy specified above, these beliefs and actions constitute a perfect Bayesian equilibrium (PBE) of the game.

A second PBE can be obtained by flipping every 0 and 1 in the strategy profile and in the corresponding beliefs; that is, the generator swaps the probabilities assigned to $(y_1, y_2)$ and the verifier swaps the outputs $(1,0)$ and $(0,1)$.

## E.2 Proof of Theorem 2

**Theorem** 2. The Markovian update schedule will converge to an $\sigma$-separated equilibrium of the Decoding Game.

### *Proof of Theorem 2.:*

We will show that the Markovian update schedule is no-regret for correct generator, and when generator receives correct signal, she will automatically perform the reversed strategy; then, if the Markovian update schedule converges to for the correct signal, it automatically satisfies that the Markovian schedule will converge for the incorrect signal and thus to a $\sigma$-separated equilibrium our Decoding Game.

$$\text{Reg}_i^{(T)} := \frac{1}{T} \left( \sum_{t=1}^{T} u_i \left( \pi_i^*, \pi_{-i}^{(t)} \right) - u_i \left( \pi_i^{(t)}, \pi_{-i}^{(t)} \right) \right)$$

and in each $t$, the utility is 1 if the order matches and 0 if not; therefore, if the schedule ensures that there exists a $T^*$, such that the period $t$ regret, defined by

$$\text{reg}_i^t = u_i \left( \pi_i^*, \pi_{-i}^{(t)} \right) - u_i \left( \pi_i^{(t)}, \pi_{-i}^{(t)} \right) = 0 \text{ for } t > T^* \tag{8}$$

Then the asymptotic regret converges to 0, and we will show that the Markovian update schedule satisfies the condition above, we will show this from generator's side and the verifier's proof follows the same steps.

The proof will follow 2 steps: firstly, we will prove a lemma that shows under Markovian schedule, once the preference relation between any two candidates, $y_1, y_2$, aligns between the generator and verifier, then they will not diverge.

**Lemma 1.** *Without loss of generality, we assume if $a_G^{(T^*)}(y_1 \mid x, s) > a_G^{(T^*)}(y_2 \mid x, s)$ and $a_V^{(T^*)}(s \mid x, y_1) > a_V^{(T^*)}(s \mid x, y_2)$, then for all $t > T^*$, under Markovian update, this preference relation stays invariant.*

This lemma is obvious as the update is the exponential of a linear combination with positive coefficients of two actions, and both exponential and linear combination preserves monotonicity.

For the second step, we will prove a lemma that ensures the Markovian update is a strict contraction mapping to ensure the preference ordering coincides in finite times.

**Lemma 2.** *Then there exists a finite round $T^*$ such that for all $t > T^*$, generator's and verifier's preference orderings coincide under the Markovian update schedule.*

***Proof of Theorem 2.:*** At round $t$ denote $g_t(y) := a_G^{(t)}(y \mid x, \text{correct})$ and $v_t(y) := a_V^{(t)}(\text{correct} \mid x, y)$ and the normalized verifier action as $\tilde{v}_t(y)$

Fix any pair of candidates $y, z$. Define the log-ratios

$$R_t^g(y, z) := \ln \frac{g_t(y)}{g_t(z)}$$

$$R_t^v(y, z) := \ln \frac{\tilde{v}_t(y)}{\tilde{v}_t(z)}$$

the generator's update is given by

$$g_{t+1}(y) \propto g_t(y)^{\alpha_t} \exp\left(\frac{1}{2}\beta_t R_t^v(y, z)\right), \quad \alpha_t := \frac{\lambda_G}{\frac{1}{\eta_G t} + \lambda_G}, \quad \beta_t := \frac{1}{\frac{1}{\eta_G t} + \lambda_G},$$

so

$$R_{t+1}^g(y, z) = \alpha_t R_t^g(y, z) + \frac{1}{2}\beta_t R_t^v(y, z). \tag{9}$$

the verifier's update is symmetric (write $\alpha_t', \beta_t'$ for the verifier's coefficients)

$$R_{t+1}^v(y, z) = \alpha_t' R_t^v(y, z) + \frac{1}{2}\beta_t' R_t^g(y, z). \tag{10}$$

Let

$$D_t(y, z) := R_t^g(y, z) - R_t^v(y, z).$$

be a signed measure of "disagreement" for the pair $y, z$, if $\lambda_G = \lambda_V, \eta_G = \eta_V$, subtract 10 from 9, get:

$$D_{t+1} = \left(\alpha_t - \frac{1}{2}\beta_t'\right) R_t^g - \left(\alpha_t' - \frac{1}{2}\beta_t\right) R_t^v = c_t D_t \tag{11}$$

where

$$c_t := \alpha_t - \frac{1}{2}\beta_t' = \frac{\lambda_G - \frac{1}{2}}{\frac{1}{\eta_G t} + \lambda_G}$$

For unequal $\lambda, \eta$ one gets two positive weights whose sum is below 1 and the same contraction argument goes through.

Assume mild regularisation: $\lambda_G, \lambda_V > \frac{1}{2}$. Then $0 < c_t < 1$ for every $t$ and $c_t < 1$, then from 11, we have

$$|D_{t+1}(y, z)| = c_t |D_t(y, z)| \implies |D_t(y, z)| \leq \left(\prod_{k=1}^{t-1} c_k\right) |D_1(y, z)| \xrightarrow[t \to \infty]{} 0$$

Hence for every pair $(y, z)$,

$$\lim_{t \to \infty} R_t^g(y, z) = \lim_{t \to \infty} R_t^v(y, z),$$

i.e. $\frac{g_t(y)}{g_t(z)} \longrightarrow \frac{\tilde{v}_t(y)}{\tilde{v}_t(z)}$ for all $y, z$. Because convergence is by a strict contraction, there exists $T_{y,z}$ such that for all $t \geq T_{y,z}$

$$\text{sign}\left(R_t^g(y, z)\right) = \text{sign}\left(R_t^v(y, z)\right)$$

so the ordering of $y$ versus $z$ is the same for generator and verifier from that round on. Since there are finitely many pairs, so

$$T^* := \max_{y \neq z} T_{y,z} < \infty$$

and for every $t \geq T^*$ the entire rankings are identical. Therefore combing Lemma 1. and Lemma 2., we verified condition 8, which shows that Markovian update schedule is indeed no-regret for the BDG.

### E.3 Proof of Proposition 1

**Proposition** 1 Under any signal distribution environment such that $\mathbf{P}(\text{correct, incorrect}) = (p, 1-p)$ s.t. $p < 1$, if the equilibrium confidence scores is conditioned that the correct candidates are greater than $\frac{1}{2}$ and the incorrect candidates is less than $\frac{1}{2}$, then the separation score is also bounded below by the same parameter in Defi.3.

$$a_V^*(\text{correct} \mid x, y_k) - a_V^*(\text{correct} \mid x, y_{k+1}) \geq \sigma_V \qquad (12)$$

**if and only if** the $\sigma_i$-separeted condition is satisfied. where $k$ is the least correct candidate and $k+1$ is the least incorrect candidate in equilibrium, determined by the candidate set cardinality and signal distribution.

*Proof of Proposition 1:* we first show that the $\sigma_i$-separatedness implies separation score bound.

for any given environment such that the signal distribution is given by $\mathbf{P}(\text{correct, incorrect} = \mathbf{P}(p, 1-p)$ this proof applies, for simplicity, we will provide the proof only for $\mathbf{P}(p, 1-p) = (0.5, 0.5)$, the only difference will be the index of the least correct and least incorrect candidate.
According to Defi. 3. the $\sigma_i$-separated condition, in equilibrium, the inequality below is satisfied

$$|a_V^*(\text{correct} \mid x, y_{\frac{n}{2}}) - a_V^*(\text{incorrect} \mid x, y_{\frac{n}{2}})| > \sigma_V$$

and

$$a_V^*(\text{correct} \mid x, y_{\frac{n}{2}}) = 1 - a_V^*(\text{incorrect} \mid x, y_{\frac{n}{2}})$$

thus the inequality becomes

$$|2 \cdot a_V^*(\text{correct} \mid x, y_{\frac{n}{2}}) - 1| > \sigma_V$$

we condition that **the equilibrium confidence for correct candidate being greater than $\frac{1}{2}$ and the incorrect candidate being less than $\frac{1}{2}$**, thus we can remove the absolute value and get $2 \cdot a_V^*(\text{correct} \mid x, y_{\frac{n}{2}}) - 1 > 0$, moreover, for $a_V^*(\text{correct} \mid x, y_{\frac{n}{2}+1})$, we have

$$1 - 2 \cdot a_V^*(\text{correct} \mid x, y_{\frac{n}{2}+1}) > \sigma_V$$

adding the two inequalities together, divided by 2, we get

$$a_V^*(\text{correct} \mid x, y_{\frac{n}{2}}) - a_V^*(\text{correct} \mid x, y_{\frac{n}{2}+1}) > \sigma_V$$

Conversely, we prove the implication in the opposite direction via contradiction: if, at termination, the $\sigma_i$–separation condition fails to hold, then the resulting separation score $S$ must satisfy

$$S \leq c < \sigma_V,$$

for some constant $c$ strictly smaller than $\sigma_V$.

We first assume that for all candidates, the correct and incorrect confidence score is bounded above uniformly by some $\sigma'$ such that $\sigma' < \sigma_V$, which is given by

$$|a_V^*(\text{correct} \mid x, y_i) - a_V^*(\text{incorrect} \mid x, y_i)| < \sigma'$$

with the boldfaced rationality condition, we have that

$$2 \cdot a_V^*(\text{correct} \mid x, y_{\frac{n}{2}}) - 1 < \sigma' \quad 1 - 2 \cdot a_V^*(\text{correct} \mid x, y_{\frac{n}{2}+1}) < \sigma'$$

adding the two equalities together we have that

$$2 \cdot a_V^*(\text{correct} \mid x, y_{\frac{n}{2}}) - 2 \cdot a_V^*(\text{correct} \mid x, y_{\frac{n}{2}+1}) < 2\sigma'$$
$$a_V^*(\text{correct} \mid x, y_{\frac{n}{2}}) - \cdot a_V^*(\text{correct} \mid x, y_{\frac{n}{2}+1}) < \sigma' < \sigma_V,$$

which showcases that without enforcing Defi.3., under the boldfaced rationality condition, the separation could be bounded above by some constant less than $\sigma_V$.

# F   Extended Related Work, Discussion, and Distinction

## F.1   Related Work

**Multi-Agent Debate Systems.** Previous work has explored mechanisms where multiple language model instances "debate" to refine and converge to a final answer [20, 12, 34, 36]. It is possible to categorize our method as a major variant of this multi-agent debate in which the interaction occurs within a game-theoretic framework, rather than directly within the language models' outputs. This structured signaling game enables BDG to enhance the correctness of outputs without relying on human feedback, by dynamically optimizing the generation and verification processes. Additionally, this approach can resolve ambiguity, confusion, and low accuracy caused by inconsistencies, but not by poor reasoning. Conventional signaling game settings have been successfully deployed for Poker [7, 8], Stratego [52], Diplomacy [21, 3, 32], and LLM tasks [27, 12]. Building on these insights, we propose a novel signaling game framework between a generator and verifier for systematic LLM output verification.

**Single-Agent Strategies.** Top-k sampling [22], nucleus sampling [30], and typical sampling [47] focus on generating high-confidence text but do not address the correctness of the outputs. Candidates were generated using these methods. Equilibrium-ranking [31] applies an average-moving strategy to the initial distribution. In contrast, BDG integrates a multistage signaling game that inherently balances correctness and consistency during the generation process. BDG can be seamlessly combined with these strategies to enhance the reliability of generated text. Furthermore, single-agent ranking is a widely used approach to select the correct output from a set of candidates generated by language models. [61] use additional human annotations to train a ranking model for response filtering. [27] trains different provers and verifiers for increasing output legibility. Although our work also utilizes existing language models as verifiers, BDG eliminates the need for additional training and does not impose specific assumptions on the structure of either the generator or verifier.

## F.2   Discussion and limitations

**Game Design over ECG.** BDG and ECG share the common goal of aligning generative models with human intentions to improve output reliability, yet they differ significantly in their game design, achieving substantial gains with reduced computational overhead. While ECG utilizes moving-average updates to foster consensus, often leading to unstable and fluctuating equilibria, BDG employs a structured Bayesian framework that drives interactions toward an optimal equilibrium with greater stability. In contrast, Prover-Verifier Games (PVGs) [27], which contribute to ChatGPT o1 [51], use a RL-based alignment and focus on adversarial training phases featured by RL and competitive dynamics. This requires intensive training and causes potential deviations from cooperative strategies.

**Robustness and Integrative Potential.** BDG achieves consistent performance improvements across diverse domains, maintaining effectiveness even with lower-quality initial LLM outputs. The framework readily integrates with existing techniques such as self-consistency and chain-of-thought prompting, while offering fast equilibrium convergence and reliable verification.

**Balancing Correctness and Reliability.** Reliability [55] tries to give an account of the prover model's failure modes and sense-making, whether the reasoning is correct or not. The resulting decoding can be arbitrarily complex [50]. In contrast, correctness allows to verify if a given solution is correct, ignoring how the generator reasoned it to be reliable (consistent with the environment). Consequently, reliability requires model outputs that are coherent and consistent to human understanding [48]. We show that it is possible to have both, without sacrificing correctness for reliability [27], and especially in high-stakes settings reliability is as important as correctness [11].

## G    From Training-free Bayesian Decoding Game (BDG) to RL-based Prover-Verifier Game(PVG)

PVG [27], structured as zero-sum games, encounter substantial challenges that undermine their efficacy in ensuring reliable outputs. The adversarial nature of zero-sum games inherently prioritizes winning over mutual consistency, which leads to strategic behavior focused on exploiting the opposing agent rather than achieving genuine correctness *e.g.,* model collapse. This often results in provers generating outputs that are optimized to mislead the verifier rather than to align with factual truth, thus producing equilibria that favor strategic manipulation over accurate assessment. Such dynamics complicate the training process, requiring extensive tuning and computational resources without guaranteeing robust, interpretable results. Furthermore, the reliance on reinforcement learning in these systems falls short of effectively replacing human feedback, as the trained verifier cannot fully replicate the nuanced judgment required to evaluate complex or ambiguous output. These limitations fall into the misalignment between training objectives and practical needs, where models become adept at adversarial optimization but lack the reliability and consistency necessary for real-world application. However, Bayesian Decoding Game (BDG) can bring the advantages of the game and bridge the purposes with proper implementations, which bypass the extensive training and adversarial pitfalls by directly modeling output verification through probabilistic reasoning, enhancing the interpretability and reliability of the generated content without the dependencies on zero-sum competition.

Here, we consider the connections between the PVG and BDG, and before the comparison, we give a brief introduction of PVG.

We consider a scenario of problems with ground-truth answers $(x, y) \in \mathcal{X} \times \mathcal{Y}$, where $x$ denotes a problem statement and $y = y(x)$ denotes the mapping from the prompt $x$ to ground-truth answer or set of answers. PVG has set the prover's type to be drawn from a known distribution, which can be either honest or sneaky, to produce an answer $z$ condition on the prompt $x$ and type; such that the honest prover is told to produce correct answers and make sure the verifier recognize the correctness, whereas the sneaky prover is told to trick the verifier by providing wrong answers that are not easy to be detected; then, after receiving solution, the verifier verifies the answer based solely on the answer and prompt, and the output $v(x, z) \in \{0, 1\}$ estimates the correctness of the solution's answer. The output of the verifier is either 1(the statement is correct) or 0(the statement is incorrect).
More precisely, The game follows the procedure below:

1. The type of prover $t$ is randomly drawn from a known distribution $P(t)$, such that $t \in \{h, s\}$, where $h, s$ dictates honest and sneaky prover's type.

2. Given a problem statement, the prover generates proofs by maximizing the payoff function(utility) of each round myopically, such that

$$z = \operatorname{argmax} r(v(x, z), \mathbb{1}_{t=h} \mathbb{1}_{\text{correct}}(x, z) + \mathbb{1}_{t=s}(1 - \mathbb{1}_{\text{correct}}(x, z)))$$

the function $r(v, q)$ satisfies that $r(1, 1) > r(0, 1) > r(1, 0) > r(0, 01)$, which corresponds to the fact that both prover prioritizing generating proofs that match their types (correct answer for honest and incorrect answer for sneaky), $\forall v \quad r(v, 1) > r(v, 0)$ and then both want their answers to be accepted as correct by the verifer $\forall q \quad r(1, q) > r(0, q)$.

3. the verifier provides verification $v \in \{0, 1\}$ based solely on the proof $x$ given that the utility is 1 if $v(x, z) = \mathbb{1}(x, z)$ and 0 otherwise
Theoretically, this is a simple Stackleberg game such that an equilibrium exists and before diving into the equilibrium strategy profile, an important definition from [27] needs to be mentioned.

**Definition 6.** *(Checkability [27]) Prover $\pi$ from model family $\mathcal{H}$ is verifiable to verifier $v$ on the data distribution $P_{XY}$ if and only if the prover always generates correct and convincing solutions, and no sneaky prover from the same model family $\mathcal{H}$ can fool the verifier into accepting incorrect solutions. In mathematical notations, the two properties are given by:*
*Completeness: $\forall x \in \mathcal{X}, \mathbb{1}_{correct}(x, \pi(x)) = v(x, \pi(x)) = 1$.*
*Soundness: $\forall \pi' \in \mathcal{H}, \forall x \in \mathcal{X}, \mathbb{1}_{correct}(x, \pi'(x)) = 0 \implies v(x, \pi'(x)) = 0$.*

With definition 6 in hand, the equilibrium strategy profile is given by *(the honest prover always provides checkable and correct proof, the sneaky prover always provides noncheckable and incorrect proof, the verifier can always verify the correctness of the given proof)*. For the neural networks to approximate the equilibrium strategies, [27] utilized a reinforcement learning-based algorithm to train the prover and the verifier.

RL-based PVG [27] can fit in part into the framework of our training-free BDG framework. As for the game-theoretic setting, PVG is a zero sum verifier-lead Stackleberg game, the strategy update schedule must be modified to fit the utility defined in [27]. Moreover, the verifier's strategy update cannot be achieved training-free as her utility only depends on the ground truth right/wrong of the candidate and thus needs to be trained; but on the prover side, both honest and sneaky prover can update strategies pain-free from the verifier's trained strategies.

Firstly, we define the strategy for verifier and prover in the same way as in BDG, such that given the environment signal, the prover generates a probability distribution for a set of answers, and the verifier always generates a probability distribution of {correct, incorrect} for each of the answers. Also, we make the same assumption that each player can observe the opponent's full strategy profile rather than the realized strategy; then, we are ready to highlight the difference in schedule update under the Markovian schedule, the condition where $v = $ correct, we will abbreviate that as correct $= \mathbf{C}$, stays the same, such that because they want to align their strategies with the verifier

$$b_{\mathrm{P}}^{(t+1)}(y \mid x, \mathbf{H}) = a_V^{(t)}(\mathbf{C} \mid x, y)$$

$$a_{\mathrm{P}}^{(t+1)}(y \mid x, \mathbf{H}) \propto \exp\left\{\frac{\frac{1}{2}b_{\mathrm{P}}^{(t+1)}(y \mid x, \mathbf{C}) + \lambda_{\mathrm{P}}\log a_{\mathrm{P}}^{(t)}(y \mid x, \mathbf{C}, b_P^{(t)})}{1/(\eta_{\mathrm{P}}t) + \lambda_{\mathrm{P}}}\right\}$$

However, for the sneaky prover, her utility is maximized when the verifier mistakens the correctness of the problem. Therefore, the optimal update schedule for the sneaky prover is given updating toward a normal distribution over the preference generated by the probability distribution of verifier's strategy. The reason for this update is because, near the correct/incorrectness boundary is where the verifier tends to make mistakes, such that

$$a_{\mathrm{P}}^{(t+1)}(y \mid x, \mathbf{S}) \propto \exp\left\{\frac{\frac{1}{2}\mathcal{N}(y \mid a_V) + \lambda_{\mathrm{P}}\log a_{\mathrm{P}}^{(t)}(y \mid x, \mathbf{I}, b_P^{(t)})}{1/(\eta_{\mathrm{P}}t) + \lambda_{\mathrm{P}}}\right\}$$

For example, if there are 10 answer candidates, the verifier's preference from her strategy is given by $y_3 \succ y_7 \succ y_6 \succ y_5 \succ y_10 \succ y_2 \succ y_9 \succ y_4 \succ y_1 \succ y_8$, then $\mathcal{N}(y \mid a_V)$ is given by

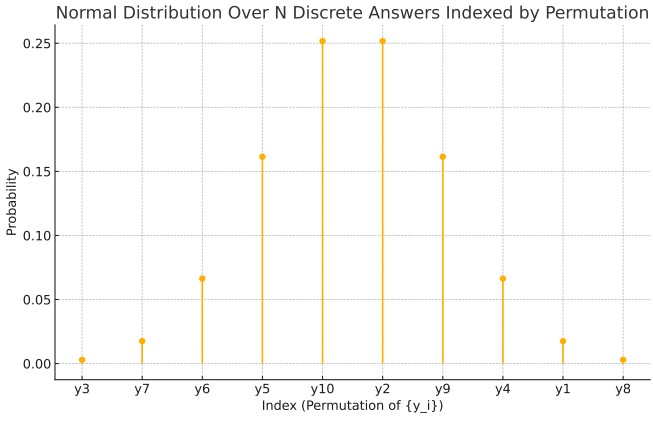

# H From Memoryless Bayesian Decoding Game (BDG) to Moving-average Equilibrium Consensus Game (ECG)

The moving average update schedule proposed by [31] requires both the generator and the verifier to keep track of the average strategy of the opponent in addition to the strategy in the last round, while our Markovian framework allows the players to be memoryless. To better compare ECG with our update schedule, we provide a general, unifying framework called the History window schedule, where the player's belief is given by the average of past history strategies for the period $n$, and at the same time, this schedule retains a large part the initial policy for each round with a stiffness parameter $\lambda_i, i \in \{G, V\}$. The belief is given by

$$
\begin{aligned}
b_{\mathrm{G}}^{(t+1)}(y \mid x, s) &= \frac{1}{n} \sum_{\tau=t-n+1}^{t} a_V^{(\tau)}(s \mid x, y) \\
b_{\mathrm{V}}^{(t+1)}(s \mid x, y) &= \frac{1}{n} \sum_{\tau=t-n+1}^{t} a_G^{(\tau)}(y \mid x, s)
\end{aligned}
\tag{13}
$$

Thus the strategy update is given by

$$
a_G^{(t+1)}(y \mid x, s) \propto \exp \left\{ \frac{\frac{1}{2} b_{\mathrm{G}}^{(t+1)}(y \mid x, s) + \lambda_{\mathrm{G}} \log a_G^{(1)}(y \mid x, s)}{1/(\eta_{\mathrm{G}} t) + \lambda_{\mathrm{G}}} \right\}
$$

$$
a_G^{(t+1)}(s \mid x, y) \propto \exp \left\{ \frac{\frac{1}{2} b_{\mathrm{V}}^{(t+1)}(y \mid x, s) + \lambda_{\mathrm{V}} \log a_V^{(1)}(s \mid x, y)}{1/(\eta_{\mathrm{V}} t) + \lambda_{\mathrm{V}}} \right\}
$$

As it can be noted in 13, if we take $n = t$, the update schedule coincides with ECG which requires the memory of the moving-average of full history, rather if we take $n = 1$, the update schedule becomes fully memoryless and requires no memory of any past events other than the last period's opponent strategy.

# I Experiment Details

**Baselines and Models.** For the fair comparision following[31], we use the same public 7B and 13B parameter models from the LLaMA family[62] and perform 16-bit inference for all our experiments. Since we have a multi-round optimization game and in order to distinguish consensus/ zero-sum games, we define ours as a verifier rather than a verifier. Across the experiments, all the approaches and orthogonal techniques involved:

- **Generative Ranking (G):** The baseline[9, 62] ranks every candidate $y$ by $P_{\text{LLM}}(y \mid x, \text{correct})$ and picks the top candidate. This is the standard approach used in past work. Due to implementation differences and non-public resources, we report the existing scores in [31].
- **verifying Ranking (D):** Following[31], this approach reweighs every query-candidate pair $(x, y)$ by $\pi_V^{(1)}(\text{correct} \mid x, y)$. Typically, this would surpass the performance of ordinary individuals, who might neglect to notice the ambiguity errors. And outstrip the generators that might trust the unreliable decoding.
- **Mutual Information Ranking (MI):** The mutual-information based baseline reweights every candidate $y$ by $P_{\text{LM}}(y \mid x, \text{correct}) \cdot P_{\text{LM}}(\text{correct} \mid x, y)$ [39].
- **Self-Contrastive Decoding (SCD):** The contrastive-based method [31, 42] utilizes the contrastive-based generator $\pi_G^{(1)}$ to reweight every candidate $y$ by $\pi_G^{(1)}(\text{correct} \mid x, y)$. This method achieves a contrasting effect by comparing negative samples instead of employing a verifier (in BDG)/ verifier (in ECG).
- **Equilibrium Consensus Verifier (ECG):** This approach is based on verifier $\pi_V^*$ [31]. It reweighs every query-candidate pair $(x, y)$ by $\pi_V^*(\text{correct} \mid x, y)$. This method, involving comprehensive policies and updates, serves as our main benchmark.
- **Bayesian Decoding Game (BDG):** This approach utilizes our Bayesian Decoding Game-based verifier $\pi_V^*$. This approach reweighs every query-candidate pair $(x, y)$ by $\pi_V^*(\text{correct} \mid x, y)$.

**Orthogonal Techniques.** Furthermore, BDG can combine chain-of-thought (CoT) [66] and few-shots setting [66] as orthogonal extra gains.

- **Chain-of-Thought (CoT):** CoT [66] prompting enables language models to generate intermediate reasoning steps, improving performance on complex tasks. By providing exemplars of reasoning chains, the model is guided to produce more coherent and accurate responses.
- **Few-Shot:** Few-shot setting [66] involves providing the model with a small number of example input-output pairs within the prompt. This technique helps the model adapt to the task at hand without additional fine-tuning, improving its ability to generalize from limited data.

**Hyperparameters.** We set $\eta_D$, $\lambda_D$ and $\eta_G$, $\lambda_G$ with 0.1 compared to ECG. Experiments are run 5000 times with early stopping based on equilibrium convergence. BDG can usually converge by 500 iterations or less. The hyperparameters can be larger according to the tasks and initial model ability.

**Extra Metrics.** Following [42], we have

- *Diversity.* This metric aggregates n-gram repetition rates:

$$\text{DIV} = \prod_{n=2}^{4} \frac{\text{unique n-grams}(x_{\text{cont}})}{\text{total n-grams}(x_{\text{cont}})}.$$

  Models that score low for diversity are prone to repetition, while models that score high for diversity are lexically diverse.
- *MAUVE.* MAUVE [53] measures the similarity between generated text and gold reference text.
- *Coherence.* [59] approximates coherence by cosine similarity between the sentence embeddings of prompt $x_{\text{pre}}$ and generated continuation $x_{\text{cont}}$:

$$\text{COH}(x_{\text{cont}}, x_{\text{pre}}) = \frac{\text{EMB}(x_{\text{pre}}) \cdot \text{EMB}(x_{\text{cont}})}{\|\text{EMB}(x_{\text{pre}})\| \cdot \|\text{EMB}(x_{\text{cont}})\|},$$

  where $\text{EMB}(x)$ represents the pre-trained SimCSE embedding [24].
- *Human Evaluation.* To further evaluate the quality of the generated text, we consider two critical aspects: *correctness* and confidence in *reliability*. More details can be found in the next section.

# J   Searching & Convergence Behavior Supplementary

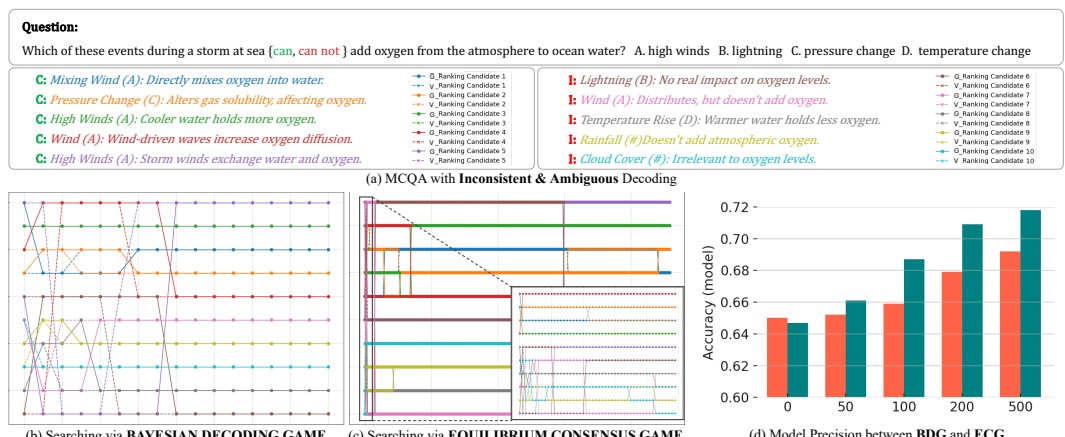

(a) MCQA with **Inconsistent & Ambiguous** Decoding

(b) Searching via **BAYESIAN DECODING GAME**      (c) Searching via **EQUILIBRIUM CONSENSUS GAME**      (d) Model Precision between **BDG** and **ECG**

Figure 8: **BDG's game design quickly reaches equilibrium and consensus between the generator and verifier, typically within 100 epochs**. In contrast, **ECG** requires significantly more epochs (3000 in this case) and exhibits continuous fluctuations (as shown in the lower right) before achieving consensus. (Zoom in for details.)

We first compare searching behaviors of BDG with the most closely related method, the ECG [31], in the multiple-choice question answering (MCQA) task [18]. Fig.8 provides a visual case study. BDG demonstrates a swift and consistent convergence in (b).

Conversely, the ECG, shown in (c), exhibits prolonged and inconsistent searching behavior. Despite continuous shifts in candidate selections, ECG fails to achieve stable convergence with persistent disagreement between the generator and verifier. (d) and Tab.5 highlights the enhanced and fast convergence properties of the BDG over the ECG.

# K   Human Evaluation

**Setting.** In this experiment, participants were tasked with evaluating the correctness of ten answers to a high-school level multiple-choice mathematics problem generated by a Large Language Model (LLM). Participants were instructed to classify each answer as correct, incorrect, or ambiguous. The experiment was conducted in two stages:

In the first stage, participants were given two minutes to classify as many answers as possible, and their results were recorded. In the second stage, participants were allowed to allocate their time freely to complete the remaining classifications, and they were asked to record the time upon completion of their classifications. Below is the questionnaire we utilized for the experiment.

Each participant was randomly assigned three distinct problems, and the corresponding solutions were classified under three conditions: without any hints, with a BDG hint, and with an ECG hint. The hints provided were rankings of the answers generated by the respective models (BDG and ECG). The assignment of different problems across the three conditions was designed to prevent memorization and to control for potential confounding effects related to the content of the specific problem. Problems were drawn from a pool of questions with similar difficulty levels, allowing for consistent observation of treatment effects across varying problem sets.

**Samples.** To better illustrate the experiment setting details, we provided the questionnaire interface, the instructions, and two cases set below.

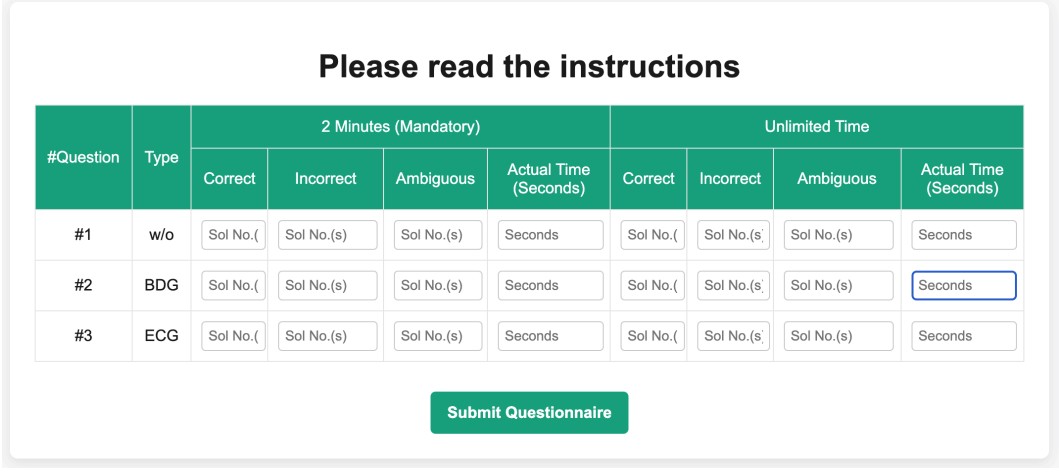

| | |
|---|---|
| **Instructions** | **For task 1**, you will judge the **correctness** of each solution under a mandatory 2-minute limit, and for the ones that you are unsure, label them as ambiguous and record the result on the answer sheet provided. If you did not finish in 2 minutes, then record your incomplete result at the 2-minute cutoff and continue your task until finished, also remember to record your completion time.

**For task 2, 3**, we will provide a reference (ranking of the solutions to assist and accelerate your correctness judgement, however, always stick to your own judgement if there exists a divergence. All other procedures are the same as task 1. |

**Question #**

*Random selected*   On a Cartesian coordinate plane, points $(1, 2)$ and $(7, 4)$ are opposite vertices of a square. What is the area of the square?

Choices   A. 20   B. 34   C. 40   D. $\sqrt{40}$

---

Task 1   *Candidates*

0.   "To find the area of the square, we multiply the length of the diagonal, which is $\sqrt{40}$, by itself. This gives us 40, so the answer is C. 40."

1.   "The distance between the points $(1, 2)$ and $(7, 4)$ is $\sqrt{40}$. This is the diagonal of the square, so the area could either be 20 or 40. The answer might be A. 20."

2.   "The diagonal of the square is $\sqrt{40}$, and the area must be equal to this value. The answer is D. $\sqrt{40}$."

3.   "First, calculate the diagonal between the points $(1, 2)$ and $(7, 4)$, which gives $\sqrt{40}$. Since this is the diagonal of the square, the area is $\frac{(\sqrt{40})^2}{2} = 20$. Therefore, the answer is A. 20."

4.   "The area of the square is found by calculating the distance between the points. The diagonal is $\sqrt{40}$, so the area is likely $\sqrt{40}$. The answer is D. $\sqrt{40}$."

5.   "First, find the distance between $(1, 2)$ and $(7, 4)$ using the distance formula: $d = \sqrt{(7-1)^2 + (4-2)^2} = \sqrt{40}$. This distance is the diagonal of the square. The area of the square is then $\frac{(\sqrt{40})^2}{2} = 20$. Therefore, the correct answer is A. 20."

6.   "Using the distance formula, the diagonal of the square is $\sqrt{40}$. Therefore, the area of the square is 20. The answer is A. 20."

7.   "First, calculate the distance between the points $(1, 2)$ and $(7, 4)$, which gives the diagonal of the square. The distance formula gives $d = \sqrt{(7-1)^2 + (4-2)^2} = \sqrt{40}$. Since the diagonal of the square is $\sqrt{40}$, the area is $(\sqrt{40})^2/2 = 20$. Therefore, the answer is A. 20."

8.   "Using the distance formula, the diagonal of the square is found to be $\sqrt{40}$. The area of the square is 20, so the answer is A. 20."

9.   "Calculate the diagonal of the square using the distance formula: $d = \sqrt{40}$. Since the diagonal is $\sqrt{40}$, the area of the square is simply 40. Therefore, the answer is C. 40."

---

**Task 2**   **BDG Guidance** [Will be hidden during evaluation]

0.

*Consistent*   "First, find the distance between $(1, 2)$ and $(7, 4)$ using the distance formula: $d = \sqrt{(7-1)^2 + (4-2)^2} = \sqrt{40}$. This distance is the diagonal of the square. The area of the square is then $\frac{(\sqrt{40})^2}{2} = 20$. Therefore, the correct answer is A. 20."

1.     "First, calculate the diagonal between the points $(1, 2)$ and $(7, 4)$, which gives $\sqrt{40}$. Since this is the diagonal of the square, the area is $\frac{(\sqrt{40})^2}{2} = 20$. Therefore, the answer is A. 20."

2.     "Using the distance formula, the diagonal of the square is $\sqrt{40}$. Therefore, the area of the square is 20. The answer is A. 20."

3.     "Using the distance formula, the diagonal of the square is found to be $\sqrt{40}$. The area of the square is 20, so the answer is A. 20."

4.     "The distance between the points $(1, 2)$ and $(7, 4)$ is $\sqrt{40}$. This is the diagonal of the square, so the area could either be 20 or 40. The answer might be A. 20."

5.     "First, calculate the distance between the points $(1, 2)$ and $(7, 4)$, which gives the diagonal of the square. The distance formula gives $d = \sqrt{(7 - 1)^2 + (4 - 2)^2} = \sqrt{40}$. Since the diagonal of the square is $\sqrt{40}$, the area is $(\sqrt{40})^2/2 = 20$. Therefore, the answer is A. 20."

6.     "The diagonal of the square is $\sqrt{40}$, and the area must be equal to this value. The answer is D. $\sqrt{40}$."

7.     "The area of the square is found by calculating the distance between the points. The diagonal is $\sqrt{40}$, so the area is likely $\sqrt{40}$. The answer is D. $\sqrt{40}$."

8.     "Calculate the diagonal of the square using the distance formula: $d = \sqrt{40}$. Since the diagonal is $\sqrt{40}$, the area of the square is simply 40. Therefore, the answer is C. 40."

9.

*Inconsistent*     "To find the area of the square, we multiply the length of the diagonal, which is $\sqrt{40}$, by itself. This gives us 40, so the answer is C. 40."

---

**Task 3**     **ECG Guidance** [Will be hidden during evaluation]

0.

*Consistent*     "First, find the distance between $(1, 2)$ and $(7, 4)$ using the distance formula: $d = \sqrt{(7 - 1)^2 + (4 - 2)^2} = \sqrt{40}$. This distance is the diagonal of the square. The area of the square is then $\frac{(\sqrt{40})^2}{2} = 20$. Therefore, the correct answer is A. 20."

1.     "Using the distance formula, the diagonal of the square is $\sqrt{40}$. Therefore, the area of the square is 20. The answer is A. 20."

2.     "First, calculate the diagonal between the points $(1, 2)$ and $(7, 4)$, which gives $\sqrt{40}$. Since this is the diagonal of the square, the area is $\frac{(\sqrt{40})^2}{2} = 20$. Therefore, the answer is A. 20."

3.     "The distance between the points $(1, 2)$ and $(7, 4)$ is $\sqrt{40}$. This is the diagonal of the square, so the area could either be 20 or 40. The answer might be A. 20."

4.     "Using the distance formula, the diagonal of the square is found to be $\sqrt{40}$. The area of the square is 20, so the answer is A. 20."

5.     "Calculate the diagonal of the square using the distance formula: $d = \sqrt{40}$. Since the diagonal is $\sqrt{40}$, the area of the square is simply 40. Therefore, the answer is C. 40."

6.     "First, calculate the distance between the points $(1, 2)$ and $(7, 4)$, which gives the diagonal of the square. The distance formula gives $d = \sqrt{(7 - 1)^2 + (4 - 2)^2} = \sqrt{40}$. Since the diagonal of the square is $\sqrt{40}$, the area is $(\sqrt{40})^2/2 = 20$. Therefore, the answer is A. 20."

7.     "The diagonal of the square is $\sqrt{40}$, and the area must be equal to this value. The answer is D. $\sqrt{40}$."

8.      "To find the area of the square, we multiply the length of the diagonal, which is $\sqrt{40}$, by itself. This gives us 40, so the answer is C. 40."

9.

*Inconsistent*      "The area of the square is found by calculating the distance between the points. The diagonal is $\sqrt{40}$, so the area is likely $\sqrt{40}$. The answer is D. $\sqrt{40}$."

---

### Question #

| | |
|---|---|
| *Random se-lected* | The standard tip in a restaurant is $15\%$ of the bill before tax. Many people in California find the tip by doubling the sales tax, which is $8.25\%$. By how many cents does this exceed the standard tip on a bill of \$60 before tax? |
| Choices | A. 90      B. 80      C. 70      D. 60 |

---

Task 1      *Candidates*

0.      "The tax is doubled to $16.5\%$, and the standard tip is $15\%$. The difference is around 70 cents for a \$60 bill, so the answer is C. 70.

1.      "To find the tip, we double the sales tax of $8.25\%$, which gives us $16.5\%$. The standard tip is $15\%$, and the difference on a \$60 bill is $60 \times (16.5\% - 15\%) = 90$ cents. Therefore, the answer is A. 90."

2.      "Doubling the sales tax of $8.25\%$ gives $16.5\%$. The difference between the standard tip and this new tip is around 60 cents for a \$60 bill. Therefore, the answer is D. 60."

3.      "Doubling the tax gives a tip of $16.5\%$, and the difference from the standard $15\%$ tip is 90 cents. Therefore, the answer is A. 90."

4.      "The standard tip is $15\%$ of 60, which is $60 \times 0.15 = 9$ dollars. Doubling the sales tax of $8.25\%$ gives $16.5\%$, and the tip based on this is $60 \times 0.165 = 9.90$ dollars. The difference is $9.90 - 9.00 = 90$ cents. Therefore, the correct answer is A. 90."

5.      "Doubling the sales tax gives $16.5\%$, and the standard tip is $15\%$. This difference gives $60 \times 0.5\% = 30$ cents. Therefore, the answer is A. 90."

6.      "The standard tip is $15\%$, or 9, and doubling the sales tax gives a tip of $16.5\%$, or 9.90. The difference is 90 cents. The correct answer is A. 90."

7.      "The difference between doubling the sales tax and using the standard tip is 90 cents for a 60 bill, but it could also be 80 cents depending on rounding. It might be A. 90 or B. 80."

8.      "The standard tip is $15\%$, and doubling the tax gives a different percentage. The difference could be either 60 or 90 cents. The answer might be D. 60."

9.      "Doubling the sales tax gives a tip rate of $16.5\%$, while the standard tip is $15\%$. The difference for a 60\$ bill is 90 cents. So the answer is A. 90."

---

**Task 2**      **BDG Guidance** [Will be hidden during evaluation]

0.

*Consistent*      "The standard tip is $15\%$ of 60, which is $60 \times 0.15 = 9$ dollars. Doubling the sales tax of $8.25\%$ gives $16.5\%$, and the tip based on this is $60 \times 0.165 = 9.90$ dollars. The difference is $9.90 - 9.00 = 90$ cents. Therefore, the correct answer is A. 90."

1.      "The standard tip is $15\%$, or 9, and doubling the sales tax gives a tip of $16.5\%$, or 9.90. The difference is 90 cents. The correct answer is A. 90."

| | |
|---|---|
| 2. | "To find the tip, we double the sales tax of $8.25\%$, which gives us $16.5\%$. The standard tip is $15\%$, and the difference on a \$60 bill is $60 \times (16.5\% - 15\%) = 90$ cents. Therefore, the answer is A. 90." |
| 3. | "Doubling the tax gives a tip of $16.5\%$, and the difference from the standard $15\%$ tip is 90 cents. Therefore, the answer is A. 90." |
| 4. | "Doubling the sales tax gives a tip rate of $16.5\%$, while the standard tip is $15\%$. The difference for a 60\$ bill is 90 cents. So the answer is A. 90." |
| 5. | "The difference between doubling the sales tax and using the standard tip is 90 cents for a 60 bill, but it could also be 80 cents depending on rounding. It might be A. 90 or B. 80." |
| 6. | "Doubling the sales tax of $8.25\%$ gives $16.5\%$. The difference between the standard tip and this new tip is around 60 cents for a \$60 bill. Therefore, the answer is D. 60." |
| 7. | "Doubling the sales tax gives $16.5\%$, and the standard tip is $15\%$. This difference gives $60 \times 0.5\% = 30$ cents. Therefore, the answer is A. 90." |
| 8. | "The tax is doubled to $16.5\%$, and the standard tip is $15\%$. The difference is around 70 cents for a \$60 bill, so the answer is C. 70. 9. |
| *Inconsistent* | "The standard tip is $15\%$, and doubling the tax gives a different percentage. The difference could be either 60 or 90 cents. The answer might be D. 60." |

| | |
|---|---|
| **Task 3** | **ECG Guidance** [Will be hidden during evaluation] |
| 0. | |
| *Consistent* | "The standard tip is $15\%$, or 9, and doubling the sales tax gives a tip of $16.5\%$, or 9.90. The difference is 90 cents. The correct answer is A. 90." |
| 1. | "The standard tip is $15\%$ of 60, which is $60 \times 0.15 = 9$ dollars. Doubling the sales tax of $8.25\%$ gives $16.5\%$, and the tip based on this is $60 \times 0.165 = 9.90$ dollars. The difference is $9.90 - 9.00 = 90$ cents. Therefore, the correct answer is A. 90." |
| 2. | "To find the tip, we double the sales tax of $8.25\%$, which gives us $16.5\%$. The standard tip is $15\%$, and the difference on a \$60 bill is $60 \times (16.5\% - 15\%) = 90$ cents. Therefore, the answer is A. 90." |
| 3. | "Doubling the tax gives a tip of $16.5\%$, and the difference from the standard $15\%$ tip is 90 cents. Therefore, the answer is A. 90." |
| 4. | "Doubling the sales tax gives a tip rate of $16.5\%$, while the standard tip is $15\%$. The difference for a 60\$ bill is 90 cents. So the answer is A. 90." |
| 5. | "The difference between doubling the sales tax and using the standard tip is 90 cents for a 60 bill, but it could also be 80 cents depending on rounding. It might be A. 90 or B. 80." |
| 6. | "Doubling the sales tax gives $16.5\%$, and the standard tip is $15\%$. This difference gives $60 \times 0.5\% = 30$ cents. Therefore, the answer is A. 90." |
| 7. | "The tax is doubled to $16.5\%$, and the standard tip is $15\%$. The difference is around 70 cents for a \$60 bill, so the answer is C. 70. |
| 8. | "The standard tip is $15\%$, and doubling the tax gives a different percentage. The difference could be either 60 or 90 cents. The answer might be D. 60." |
| 9. | |
| *Inconsistent* | "Doubling the sales tax of $8.25\%$ gives $16.5\%$. The difference between the standard tip and this new tip is around 60 cents for a \$60 bill. Therefore, the answer is D. 60." |

**Results.** To differentiate between expert and non-expert participants, a threshold of 150 seconds was set based on empirical observations of participant behavior. This threshold was corroborated by a scatterplot that visually demonstrated the partitioning between experts and non-experts, supporting the appropriateness of the selected cut-off time for classification performance. From the 183 samples we collected, we have come to conclusions:

Firstly, human evaluation on these LLM-generated solutions have instrinsic unreliability and ambiguity, especially for non-expert participants; under the 2-minute time constraint, there were 28 participants who scored an accuracy of 50% or below, even without time constraint, there were 10 participants in total whose accuracies were 60% or below.

Secondly, with the hint of BDG, we observed that the participants in both the expert and the non-expert groups outperformed the participants without the hint or the ECG hint; more importantly, with the hint of BDG, the performance of the non-expert participants improved significantly with respect to the expert participants, whereas the ECG hint did not achieve this.

Thirdly, across problems, samples with BDG hint achieved the highest accuracies for 7/10 problems and fastest completion also for 7/10 problems. This result showcases the robustness of BDG hint also within the same problems.

Lastly, the BCG hint has also helped non-expert participants spot ambiguous solutions more accurately compared to non-expert participamts with the ECG hint or without the hint.

**Generations.**

---

**Question.**
Suppose the graph of $y = f(x)$ includes the points $(1, 5)$, $(2, 3)$, and $(3, 1)$. Based only on this information, there are two points that must be on the graph of $y = f(f(x))$. If we call those points $(a, b)$ and $(c, d)$, what is $ab + cd$?

**BDG.**
"Given the points $(1, 5), (2, 3)$, and $(3, 1)$ on the graph of $y = f(x)$, we compute $f(f(2)) = f(3) = 1$ and $f(f(3)) = f(1) = 5$. So the points $(2, 1)$ and $(3, 5)$ are on the graph of $y = f(f(x))$. Therefore, $ab + cd = 2 \times 1 + 3 \times 5 = 17$. "

**ECG.**
"From $f(x)$, we see that $f(f(2)) = 1$ and $f(f(3)) = 5$. Therefore, $ab + cd = 17$. The answer is 17"

---

**Question.**
Positive integers $x$ and $y$ have a product of 56 and $x < y$. Seven times the reciprocal of the smaller integer plus 14 times the reciprocal of the larger integer equals 4. What is the value of $x$?

**BDG.**
"Let the integers be $x$ and $y$. Since $x \cdot y = 56$ and $x < y$, try small values. Testing $x = 2$ and $y = 28$ satisfies $7 \cdot \frac{1}{x} + 14 \cdot \frac{1}{y} = 4$. Therefore, $x = 2$, so the answer is 2."

**ECG.**
"Since $x \cdot y = 56$, try small values of $x$ like 2. This satisfies the equation, so the answer is 2."

---

**Question.**
Six positive integers are written on the faces of a cube. Each vertex is labeled with the product of the three numbers on the faces adjacent to the vertex. If the sum of the numbers on the vertices is equal to 1001, then what is the sum of the numbers written on the faces?

**BDG.**
"The numbers on the faces of the cube contribute to the product at each vertex. Given that the sum of the vertex products is 1001, and knowing the cube's geometry, we deduce that the sum of the numbers on the faces is 31."

**ECG.**
"Using the fact that the sum of the vertex products is 1001, and knowing there are six faces, we calculate that the sum of the face numbers is 31. So the answer is 31."

# NeurIPS Paper Checklist

1. **Claims**

   Question: Do the main claims made in the abstract and introduction accurately reflect the paper's contributions and scope?

   Answer: [Yes]

   Justification: The abstract and introduction clearly state our paper's main contributions: introducing a novel game-theoretic approach (Bayesian Decoding Game) that enhances consistency and reliability during LLM decoding. The claims about performance improvements (e.g., 78.1 LLaMA13B vs 76.6 PaLM540B) are supported by our experimental results in Section 3.

   Guidelines:

   - The answer NA means that the abstract and introduction do not include the claims made in the paper.
   - The abstract and/or introduction should clearly state the claims made, including the contributions made in the paper and important assumptions and limitations. A No or NA answer to this question will not be perceived well by the reviewers.
   - The claims made should match theoretical and experimental results, and reflect how much the results can be expected to generalize to other settings.
   - It is fine to include aspirational goals as motivation as long as it is clear that these goals are not attained by the paper.

2. **Limitations**

   Question: Does the paper discuss the limitations of the work performed by the authors?

   Answer: [Yes]

   Justification: We discuss limitations in Section 4 (Discussion and limitations), noting that our approach requires explicit specification of correctness consistency branches during the game process, and that this alignment is primarily intended to match human intent with model outputs. We also acknowledge potential for further improvement by adding multi-metrics and multiple agents to achieve game-based deliberation.

   Guidelines:

   - The answer NA means that the paper has no limitation while the answer No means that the paper has limitations, but those are not discussed in the paper.
   - The authors are encouraged to create a separate "Limitations" section in their paper.
   - The paper should point out any strong assumptions and how robust the results are to violations of these assumptions (e.g., independence assumptions, noiseless settings, model well-specification, asymptotic approximations only holding locally). The authors should reflect on how these assumptions might be violated in practice and what the implications would be.
   - The authors should reflect on the scope of the claims made, e.g., if the approach was only tested on a few datasets or with a few runs. In general, empirical results often depend on implicit assumptions, which should be articulated.
   - The authors should reflect on the factors that influence the performance of the approach. For example, a facial recognition algorithm may perform poorly when image resolution is low or images are taken in low lighting. Or a speech-to-text system might not be used reliably to provide closed captions for online lectures because it fails to handle technical jargon.
   - The authors should discuss the computational efficiency of the proposed algorithms and how they scale with dataset size.
   - If applicable, the authors should discuss possible limitations of their approach to address problems of privacy and fairness.
   - While the authors might fear that complete honesty about limitations might be used by reviewers as grounds for rejection, a worse outcome might be that reviewers discover limitations that aren't acknowledged in the paper. The authors should use their best judgment and recognize that individual actions in favor of transparency play an important role in developing norms that preserve the integrity of the community. Reviewers will be specifically instructed to not penalize honesty concerning limitations.

3. **Theory assumptions and proofs**

   Question: For each theoretical result, does the paper provide the full set of assumptions and a complete (and correct) proof?

   Answer: [Yes]

   Justification: The paper provides complete theoretical foundations and proofs for all theoretical claims. Theorems 1, 2, and Proposition 1 are stated with clear assumptions and are accompanied by rigorous proofs in Appendix sections A.3 (App_thm1), A.4 (App_thm3), and A.5 (App_prop1).

   Guidelines:

   - The answer NA means that the paper does not include theoretical results.
   - All the theorems, formulas, and proofs in the paper should be numbered and cross-referenced.
   - All assumptions should be clearly stated or referenced in the statement of any theorems.
   - The proofs can either appear in the main paper or the supplemental material, but if they appear in the supplemental material, the authors are encouraged to provide a short proof sketch to provide intuition.
   - Inversely, any informal proof provided in the core of the paper should be complemented by formal proofs provided in appendix or supplemental material.
   - Theorems and Lemmas that the proof relies upon should be properly referenced.

4. **Experimental result reproducibility**

   Question: Does the paper fully disclose all the information needed to reproduce the main experimental results of the paper to the extent that it affects the main claims and/or conclusions of the paper (regardless of whether the code and data are provided or not)?

   Answer: [Yes]

   Justification: Section 3 provides detailed experimental setups, datasets used (MMLU, ARC-Easy, ARC-Challenge, RACE-High, etc.), model specifications (LLaMA-7B/13B), and implementation details. Additionally, we include a detailed reproducibility statement in Appendix Section 1 with pseudocode implementation (Algorithm 1) of the Bayesian Decoding Game.

   Guidelines:

   - The answer NA means that the paper does not include experiments.
   - If the paper includes experiments, a No answer to this question will not be perceived well by the reviewers: Making the paper reproducible is important, regardless of whether the code and data are provided or not.
   - If the contribution is a dataset and/or model, the authors should describe the steps taken to make their results reproducible or verifiable.
   - Depending on the contribution, reproducibility can be accomplished in various ways. For example, if the contribution is a novel architecture, describing the architecture fully might suffice, or if the contribution is a specific model and empirical evaluation, it may be necessary to either make it possible for others to replicate the model with the same dataset, or provide access to the model. In general. releasing code and data is often one good way to accomplish this, but reproducibility can also be provided via detailed instructions for how to replicate the results, access to a hosted model (e.g., in the case of a large language model), releasing of a model checkpoint, or other means that are appropriate to the research performed.
   - While NeurIPS does not require releasing code, the conference does require all submissions to provide some reasonable avenue for reproducibility, which may depend on the nature of the contribution. For example
     (a) If the contribution is primarily a new algorithm, the paper should make it clear how to reproduce that algorithm.
     (b) If the contribution is primarily a new model architecture, the paper should describe the architecture clearly and fully.
     (c) If the contribution is a new model (e.g., a large language model), then there should either be a way to access this model for reproducing the results or a way to reproduce the model (e.g., with an open-source dataset or instructions for how to construct the dataset).

(d) We recognize that reproducibility may be tricky in some cases, in which case authors are welcome to describe the particular way they provide for reproducibility. In the case of closed-source models, it may be that access to the model is limited in some way (e.g., to registered users), but it should be possible for other researchers to have some path to reproducing or verifying the results.

5. **Open access to data and code**

Question: Does the paper provide open access to the data and code, with sufficient instructions to faithfully reproduce the main experimental results, as described in supplemental material?

Answer: [Yes]

Justification: We provide model-agnostic pseudocode implementation in Algorithm 1 (Appendix Section 1) and detailed instructions for reproducing our results. The benchmarks we used (MMLU, ARC, RACE, etc.) are publicly available datasets, and we used open-source LLaMA models for our experiments. Complete implementation details are provided in Appendix Sections 1 and 7.

Guidelines:

- The answer NA means that paper does not include experiments requiring code.
- Please see the NeurIPS code and data submission guidelines (`https://nips.cc/public/guides/CodeSubmissionPolicy`) for more details.
- While we encourage the release of code and data, we understand that this might not be possible, so "No" is an acceptable answer. Papers cannot be rejected simply for not including code, unless this is central to the contribution (e.g., for a new open-source benchmark).
- The instructions should contain the exact command and environment needed to run to reproduce the results. See the NeurIPS code and data submission guidelines (`https://nips.cc/public/guides/CodeSubmissionPolicy`) for more details.
- The authors should provide instructions on data access and preparation, including how to access the raw data, preprocessed data, intermediate data, and generated data, etc.
- The authors should provide scripts to reproduce all experimental results for the new proposed method and baselines. If only a subset of experiments are reproducible, they should state which ones are omitted from the script and why.
- At submission time, to preserve anonymity, the authors should release anonymized versions (if applicable).
- Providing as much information as possible in supplemental material (appended to the paper) is recommended, but including URLs to data and code is permitted.

6. **Experimental setting/details**

Question: Does the paper specify all the training and test details (e.g., data splits, hyperparameters, how they were chosen, type of optimizer, etc.) necessary to understand the results?

Answer: [Yes]

Justification: Section 3 and Appendix Section 7 provide comprehensive details of our experimental settings, including hyperparameters ($\eta_D$, $\lambda_D$, $\eta_G$, $\lambda_G$ set to 0.1), model specifications (LLaMA 7B and 13B with 16-bit inference), datasets used, and evaluation metrics. We also specify hardware configurations (NVIDIA A6000 and A100 GPUs) and runtime information.

Guidelines:

- The answer NA means that the paper does not include experiments.
- The experimental setting should be presented in the core of the paper to a level of detail that is necessary to appreciate the results and make sense of them.
- The full details can be provided either with the code, in appendix, or as supplemental material.

7. **Experiment statistical significance**

Question: Does the paper report error bars suitably and correctly defined or other appropriate information about the statistical significance of the experiments?

Answer: [Yes]

Justification: Our experiments report statistical significance through comprehensive evaluations across multiple datasets and models. For our human evaluation (Section 3.1 and Appendix Section 9), we collected 183 samples with clear statistical differences between conditions. We ran our game-theoretic experiments 5000 times with early stopping based on equilibrium convergence to ensure robustness of results.

Guidelines:

- The answer NA means that the paper does not include experiments.
- The authors should answer "Yes" if the results are accompanied by error bars, confidence intervals, or statistical significance tests, at least for the experiments that support the main claims of the paper.
- The factors of variability that the error bars are capturing should be clearly stated (for example, train/test split, initialization, random drawing of some parameter, or overall run with given experimental conditions).
- The method for calculating the error bars should be explained (closed form formula, call to a library function, bootstrap, etc.)
- The assumptions made should be given (e.g., Normally distributed errors).
- It should be clear whether the error bar is the standard deviation or the standard error of the mean.
- It is OK to report 1-sigma error bars, but one should state it. The authors should preferably report a 2-sigma error bar than state that they have a 96% CI, if the hypothesis of Normality of errors is not verified.
- For asymmetric distributions, the authors should be careful not to show in tables or figures symmetric error bars that would yield results that are out of range (e.g. negative error rates).
- If error bars are reported in tables or plots, The authors should explain in the text how they were calculated and reference the corresponding figures or tables in the text.

8. **Experiments compute resources**

Question: For each experiment, does the paper provide sufficient information on the computer resources (type of compute workers, memory, time of execution) needed to reproduce the experiments?

Answer: [Yes]

Justification: In the Reproducibility Statement (Appendix Section 1), we specify that experiments were conducted on NVIDIA A6000 and A100 GPUs, with runtimes ranging from 0.5 to 6 hours depending on model size, task, and experimental settings. We also note that BDG typically converges within 500 iterations or less, providing a clear indication of computational requirements.

Guidelines:

- The answer NA means that the paper does not include experiments.
- The paper should indicate the type of compute workers CPU or GPU, internal cluster, or cloud provider, including relevant memory and storage.
- The paper should provide the amount of compute required for each of the individual experimental runs as well as estimate the total compute.
- The paper should disclose whether the full research project required more compute than the experiments reported in the paper (e.g., preliminary or failed experiments that didn't make it into the paper).

9. **Code of ethics**

Question: Does the research conducted in the paper conform, in every respect, with the NeurIPS Code of Ethics https://neurips.cc/public/EthicsGuidelines?

Answer: [Yes]

Justification: Our research fully conforms to the NeurIPS Code of Ethics. We have used publicly available models and datasets, appropriately credited prior work, and we discuss potential ethical risks and societal impact in Appendix Section 2, including considerations about potential for misuse in disinformation scenarios.

Guidelines:

- The answer NA means that the authors have not reviewed the NeurIPS Code of Ethics.
- If the authors answer No, they should explain the special circumstances that require a deviation from the Code of Ethics.
- The authors should make sure to preserve anonymity (e.g., if there is a special consideration due to laws or regulations in their jurisdiction).

10. **Broader impacts**

Question: Does the paper discuss both potential positive societal impacts and negative societal impacts of the work performed?

Answer: [Yes]

Justification: We discuss broader impacts in Appendix Section 2 (Potential Ethics Risks and Societal Impact). Positive impacts include improved reliability and consistency of LLM outputs, which enhances trustworthiness. We also acknowledge potential negative impacts such as more convincing disinformation if the techniques are misused, as illustrated in Figure 9.

Guidelines:

- The answer NA means that there is no societal impact of the work performed.
- If the authors answer NA or No, they should explain why their work has no societal impact or why the paper does not address societal impact.
- Examples of negative societal impacts include potential malicious or unintended uses (e.g., disinformation, generating fake profiles, surveillance), fairness considerations (e.g., deployment of technologies that could make decisions that unfairly impact specific groups), privacy considerations, and security considerations.
- The conference expects that many papers will be foundational research and not tied to particular applications, let alone deployments. However, if there is a direct path to any negative applications, the authors should point it out. For example, it is legitimate to point out that an improvement in the quality of generative models could be used to generate deepfakes for disinformation. On the other hand, it is not needed to point out that a generic algorithm for optimizing neural networks could enable people to train models that generate Deepfakes faster.
- The authors should consider possible harms that could arise when the technology is being used as intended and functioning correctly, harms that could arise when the technology is being used as intended but gives incorrect results, and harms following from (intentional or unintentional) misuse of the technology.
- If there are negative societal impacts, the authors could also discuss possible mitigation strategies (e.g., gated release of models, providing defenses in addition to attacks, mechanisms for monitoring misuse, mechanisms to monitor how a system learns from feedback over time, improving the efficiency and accessibility of ML).

11. **Safeguards**

Question: Does the paper describe safeguards that have been put in place for responsible release of data or models that have a high risk for misuse (e.g., pretrained language models, image generators, or scraped datasets)?

Answer: [NA]

Justification: Our paper introduces a decoding methodology rather than releasing new models or datasets. We utilize existing publicly available models (LLaMA) and benchmark datasets. Our method aims to improve reliability and reduce misinformation, serving as a safeguard itself rather than creating new risks that require additional protections.

Guidelines:

- The answer NA means that the paper poses no such risks.
- Released models that have a high risk for misuse or dual-use should be released with necessary safeguards to allow for controlled use of the model, for example by requiring that users adhere to usage guidelines or restrictions to access the model or implementing safety filters.
- Datasets that have been scraped from the Internet could pose safety risks. The authors should describe how they avoided releasing unsafe images.

- We recognize that providing effective safeguards is challenging, and many papers do not require this, but we encourage authors to take this into account and make a best faith effort.

12. **Licenses for existing assets**

    Question: Are the creators or original owners of assets (e.g., code, data, models), used in the paper, properly credited and are the license and terms of use explicitly mentioned and properly respected?

    Answer: [Yes]

    Justification: We properly credit all existing assets used in our research. We cite the original papers for LLaMA, deepseek models and all benchmark datasets. All models and datasets used are publicly available resources with appropriate licenses for research purposes.

    Guidelines:

    - The answer NA means that the paper does not use existing assets.
    - The authors should cite the original paper that produced the code package or dataset.
    - The authors should state which version of the asset is used and, if possible, include a URL.
    - The name of the license (e.g., CC-BY 4.0) should be included for each asset.
    - For scraped data from a particular source (e.g., website), the copyright and terms of service of that source should be provided.
    - If assets are released, the license, copyright information, and terms of use in the package should be provided. For popular datasets, `paperswithcode.com/datasets` has curated licenses for some datasets. Their licensing guide can help determine the license of a dataset.
    - For existing datasets that are re-packaged, both the original license and the license of the derived asset (if it has changed) should be provided.
    - If this information is not available online, the authors are encouraged to reach out to the asset's creators.

13. **New assets**

    Question: Are new assets introduced in the paper well documented and is the documentation provided alongside the assets?

    Answer: [Yes]

    Justification: The primary new asset introduced in our paper is the Bayesian Decoding Game (BDG) framework. This is thoroughly documented through formal definitions, pseudocode (Algorithm 1 in Appendix Section 1), and detailed explanations of the game-theoretic mechanisms. We provide comprehensive implementation details that allow for reproduction of our approach.

    Guidelines:

    - The answer NA means that the paper does not release new assets.
    - Researchers should communicate the details of the dataset/code/model as part of their submissions via structured templates. This includes details about training, license, limitations, etc.
    - The paper should discuss whether and how consent was obtained from people whose asset is used.
    - At submission time, remember to anonymize your assets (if applicable). You can either create an anonymized URL or include an anonymized zip file.

14. **Crowdsourcing and research with human subjects**

    Question: For crowdsourcing experiments and research with human subjects, does the paper include the full text of instructions given to participants and screenshots, if applicable, as well as details about compensation (if any)?

    Answer: [Yes]

    Justification: Our paper includes a user study with 183 participants. Appendix Section 9 (Human Evaluation) provides the complete experimental setup, including instructions given to participants, example questionnaires, and screenshots of the interface used. We report detailed analysis of the results including methodology for expert/non-expert classification.

    Guidelines:

- The answer NA means that the paper does not involve crowdsourcing nor research with human subjects.
- Including this information in the supplemental material is fine, but if the main contribution of the paper involves human subjects, then as much detail as possible should be included in the main paper.
- According to the NeurIPS Code of Ethics, workers involved in data collection, curation, or other labor should be paid at least the minimum wage in the country of the data collector.

15. **Institutional review board (IRB) approvals or equivalent for research with human subjects**

Question: Does the paper describe potential risks incurred by study participants, whether such risks were disclosed to the subjects, and whether Institutional Review Board (IRB) approvals (or an equivalent approval/review based on the requirements of your country or institution) were obtained?

Answer: [Yes]

Justification: Our human evaluation study involved minimal risk to participants as they were only evaluating mathematical solutions. The study was conducted in accordance with our institution's ethical guidelines for human subject research, with appropriate informed consent from all participants. The task involved no collection of personal or sensitive information.

Guidelines:

- The answer NA means that the paper does not involve crowdsourcing nor research with human subjects.
- Depending on the country in which research is conducted, IRB approval (or equivalent) may be required for any human subjects research. If you obtained IRB approval, you should clearly state this in the paper.
- We recognize that the procedures for this may vary significantly between institutions and locations, and we expect authors to adhere to the NeurIPS Code of Ethics and the guidelines for their institution.
- For initial submissions, do not include any information that would break anonymity (if applicable), such as the institution conducting the review.

16. **Declaration of LLM usage**

Question: Does the paper describe the usage of LLMs if it is an important, original, or non-standard component of the core methods in this research? Note that if the LLM is used only for writing, editing, or formatting purposes and does not impact the core methodology, scientific rigorousness, or originality of the research, declaration is not required.

Answer: [Yes]

Justification: Our research focuses directly on improving LLM decoding methods, and we use LLaMA models (7B and 13B) as the foundation for our experiments. Section 3 clearly describes how these models were utilized in our Bayesian Decoding Game framework, including prompting methods, model configurations, and implementation details.

Guidelines:

- The answer NA means that the core method development in this research does not involve LLMs as any important, original, or non-standard components.
- Please refer to our LLM policy (`https://neurips.cc/Conferences/2025/LLM`) for what should or should not be described.

