# OpenReview forum: "From Self-Check to Consensus: Bayesian Strategic Decoding in Large Language Models"
_NeurIPS.cc/2025/Conference — NeurIPS 2025 poster_

### Official Review · Reviewer_sYwu · 2025-06-28

**Clarity:** 2
**Significance:** 3
**Originality:** 3
**Rating:** 4
**Confidence:** 2

**Summary:**

The manuscript titled "From Self-Check to Consensus: Bayesian Strategic Decoding in Large Language Models" addresses the prevalent issue of logical inconsistency in Large Language Models (LLMs) during multi-turn inference processes. The authors identify that existing approaches, such as single-agent reflection and multi-agent debate, often compromise accuracy in favor of consistency. To overcome these limitations, the paper introduces a novel game-theoretic consensus mechanism named the Bayesian Decoding Game (BDG). This framework models the decoding process as a multistage Bayesian Decoding Game, facilitating strategic interactions between a Generator and a Verifier to achieve consensus on reliable outputs. The proposed method is model-agnostic, demonstrating that smaller models can outperform significantly larger counterparts without additional training. Experimental results across various benchmarks, including MMLU, ARC, and RACE, underscore the effectiveness of BDG in enhancing both consistency and correctness of LLM outputs. Additionally, a user study highlights BDG's potential in bridging the expertise gap by improving performance for both experts and non-experts.

**Questions:**

1. How sensitive is BDG to the choice of hyperparameters such as the learning rates (η) and stiffness parameters (γ)? Have the authors conducted any ablation studies to understand the impact of these parameters on performance?

2. The paper mentions the possibility of incorporating multi-metrics and multiple agents to achieve game-based deliberation. Have the authors explored any preliminary results in this direction, and what challenges do they anticipate?

3. Can BDG be efficiently integrated into real-time applications where rapid decoding is essential? If so, what optimizations are necessary to achieve this?

**Ethical Concerns:**

["NO or VERY MINOR ethics concerns only"]

**Final Justification:**

The authors responded to my concerns about computational overhead and broader comparison. Although they did not fully address my concern, I still hold a positive attitude toward this paper. So I decided to maintain my original positive rating.

**Limitations:**

There is no potential negative societal impact.

**Quality:**

2

**Strengths And Weaknesses:**

Strengths:

1. The introduction of the Bayesian Decoding Game (BDG) as a game-theoretic framework is a novel contribution that effectively addresses the limitations of existing self-verification methods in LLMs. By framing the decoding process as a signaling game, the authors provide a structured mechanism for LLMs to self-check and reach consensus without relying heavily on human feedback.

2. The paper offers a comprehensive theoretical foundation for BDG, including the formulation of the decoding process, the concept of Separating Equilibrium (SE), and the implementation of no-regret optimization through Markovian strategy updates. Theorems and propositions are clearly stated, demonstrating the framework's ability to prevent collusion and ensure reliable behavior.

3. Extensive experiments across multiple benchmarks (e.g., MMLU, ARC-E, ARC-C, RACE-H) convincingly demonstrate BDG's superiority over baseline models and other game-theoretic methods like ECG. Notably, the ability of smaller models (e.g., LLaMA13B) to outperform larger models (e.g., PaLM540B) without additional training is a significant achievement.


Weaknesses

1. While the theoretical aspects of BDG are well-elaborated, certain sections of the manuscript are dense and may be challenging for readers unfamiliar with game theory or advanced machine learning concepts. Enhanced explanations or illustrative examples could improve accessibility.

2. While BDG demonstrates improved convergence rates compared to ECG, the paper does not extensively discuss the computational overhead or scalability of BDG in extremely large-scale deployments. Insights into resource requirements and potential bottlenecks would be beneficial.

3. The empirical evaluation primarily focuses on comparisons with ECG and baseline generative rankings. Including a wider range of existing self-verification and consistency-enhancing methods could further strengthen the evidence for BDG's superiority.

---

> ### Author Rebuttal · Authors · 2025-07-28
>
> **We thank Reviewer sYwu for their positive evaluation of BDG’s theoretical contributions, its equilibrium structure, and the consistent empirical gains observed across strong baselines. Below we clarify several points raised, with emphasis on our design intentions and implementation choices.**
>
> ---
>
> **1. On theoretical density and accessibility**
>
> We appreciate the concern regarding theoretical accessibility. While §2.2–§2.3 introduces the core BDG framework formally, we emphasize that the manuscript integrates illustrative guidance throughout:
>
> - We provide intuition-first examples in §2.1 and Figure 1, as well as visual teasers in Figures 2a–2b to concretely motivate the generator–verifier game dynamics before formalizing the model.
> - Our narrative uses example-driven hooks across sections (e.g., policy ordering examples, belief mismatches) that directly correspond to the components of the signaling game.
> - Appendix I includes annotated pseudocode, and internal equation cross-references are used throughout to improve navigability between definition and application.
>
> Other reviewers (e.g., HSTc, SV6t) found the theoretical formulation rigorous and internally consistent. Our planned edits will preserve this formal integrity while expanding accessibility, for example, by relocating detailed proofs to the appendix, adding a notation table, and expanding comparative commentary on related frameworks (e.g., ECG, USC). These changes aim not to simplify the theory, but to better structure the entry points for different reader backgrounds.
>
> **2. On computational scalability and overhead**
>
> BDG was designed to operate with minimal decoding overhead. Rather than introducing new sampling or training steps, the generator and verifier engage in lightweight belief updates over a fixed candidate set. As described in §3.1 and Appendix I:
>
> - Candidate generation occurs once per query; all subsequent steps involve only softmax-based strategy updates (Eq. 4–6), using internal reweighting rather than additional model calls.
> - Generator and verifier operate in parallel and update synchronously, allowing for batch-efficient and hardware-parallelizable implementation.
> - Convergence is typically reached within 50–100 steps (Fig. 3a), and policy entropy stabilizes early (Fig. 4a), reducing unnecessary computation.
>
> We acknowledge that the original version focused more on algorithmic detail than runtime profiling. For completeness, we will include decoding-call counts in the final appendix, and contrast BDG with sampling-heavy baselines like Self-Consistency. However, we stress that BDG’s inference-time design is already efficient by construction.
>
> **3. On broader comparison to self-verification and consistency methods**
>
> Our baselines include a representative spectrum of decoding-time techniques:
>
> - **Verifier-based ranking (D)**, **Mutual Information reweighting (MI)**, and **Self-Contrastive Decoding (SCD)**. These are all evaluated under the same budget (§3.1, Appendix I).
> - **ECG** [31], the most relevant game-theoretic baseline, is thoroughly compared in terms of convergence behavior (Fig. 3), entropy stability (Fig. 4), and disagreement rates (Table 2).
> - BDG consistently outperforms all baselines, even enabling smaller models like LLaMA13B to rival or exceed the performance of PaLM540B (Table 1).
>
> Our focus is on decoding-time strategies with no retraining and no auxiliary supervision. While training-time methods (e.g., deliberation fine-tuning, feedback alignment) exist, they lie outside the inference-time scope of BDG and would introduce orthogonal cost tradeoffs. That said, we will expand §4.1 to include a broader landscape comparison and discuss how BDG complements other approaches and vice versa.
>
> ---
>
> We appreciate the reviewer’s constructive suggestions and will revise the paper accordingly to improve clarity, expand evaluation context, and quantify inference cost. BDG provides a theoretically grounded, practically viable decoding strategy that addresses both correctness and consistency, bridging a critical gap in LLM self-verification.

---

> > ### Comment · Reviewer_sYwu · 2025-08-05
> >
> > Thanks for the response. I decide to maintain my positive rating.

---

### Official Review · Reviewer_RFvm · 2025-07-01

**Clarity:** 3
**Significance:** 2
**Originality:** 3
**Rating:** 4
**Confidence:** 3

**Summary:**

This work addresses the problem of logical inconsistency across multi-turn inference processes in LLMs. Since existing approaches like RLHF have limitations, the authors propose BDG, a novel game-theoretic consensus mechanism. The method models the decoding process as a multistage Bayesian Decoding Game with two LLMs: a generator and a verifier. The generator ranks candidate answers using strategy probabilities, while the verifier assesses each candidate with correctness judgment probabilities. The goal is to achieve reliability, defined as simultaneous consistency and correctness, by aligning these two rankings. The generator and verifier engage in strategic interactions, observing each other's strategies and updating them through Markovian strategy updates until they converge to the same ranking. However, multiple Nash equilibria exist, creating risks of collusive reinforcement where agents converge on shared errors. To prevent this collusion, BDG introduces the σi-separated equilibrium condition, which enforces strategic separation by maintaining a minimum distance between correct and incorrect candidate judgment probabilities. Through no-regret optimization with Markovian updates, the game is mathematically guaranteed to converge to an optimal σi-separated equilibrium (Theorm 2), ensuring both consistency and correctness while preventing collusive behaviors.

**Questions:**

- Experiments use LLaMA, which seems outdated. Is there a reason for not using LLaMA-2 or 3?

> (L198) For open-ended generative tasks, we construct Y by sampling candidates from the LLM’s distribution P_LLM(y | q, correct) using nucleus [30] and top-k [22] sampling methods
- Is y here a single token or can it be a token sequence?

**Ethical Concerns:**

["NO or VERY MINOR ethics concerns only"]

**Final Justification:**

Most of the concerns have been addressed in the rebuttal. Regarding the concern about outdated models, since the current evaluation is limited to relatively old models, I maintain my original score rather than raise it.

**Limitations:**

yes

**Quality:**

3

**Strengths And Weaknesses:**

- The paper is relatively well-structured
- Performance improvements are substantial
- Multiple iterations are required to decode a single answer. For example, Figure 3 shows BDG requires around 100 iterations for generator and verifier rankings to achieve clear separation and alignment. A comparison of inference counts with standard decoding strategies would strengthen the argument
- Nearly all experimental tasks are limited to multiple-choice problems. GSM8K outputs numerical values with exact matching, lacking the feel of open-ended generation. Evaluation on tasks like code generation, machine translation, or summarization appears necessary

---

> ### Author Rebuttal · Authors · 2025-07-28
>
> **We thank Reviewer RFvm for recognizing the clear structure of our work and the substantial improvements of BDG across tasks. Below, we respond to the specific concerns regarding inference cost and the range of evaluated tasks.**
>
> ---
>
> **1. On iteration count and inference efficiency**
>
> We agree that decoding efficiency is important for practical deployment and AI trustworthiness. While BDG introduces an iterative refinement process between generator and verifier, our framework is designed for rapid convergence:
>
> - In Figure 3a, convergence is typically achieved in fewer than 100 iterations, and often within 50 steps. Unlike methods that require sampling hundreds of completions (e.g., self-consistency), BDG deterministically reweights a fixed candidate set without additional calls to the language model after candidate generation (§3.2, L239–L246).
> - Importantly, the generator and verifier can be run in parallel during each update step, as their beliefs are updated synchronously (Eq. 4–6). Thus, wall-clock latency remains modest.
> - The policy entropy trajectory (Fig. 4a) shows that BDG stabilizes significantly faster than prior multi-agent methods like ECG, which exhibit oscillatory behavior and longer plateaus before convergence (Fig. 3b). BDG’s convergence criteria (Definition 3) also allow early stopping once preference alignment and separation are achieved.
>
> We will clarify these design considerations in the final appendix.
>
> **2. On limited coverage of open-ended generation tasks**
>
> Our focus in this work is on structured and semi-structured QA tasks (e.g., MMLU, ARC, RACE, PubMedQA) that allow controlled evaluation of correctness and consistency. We agree that evaluating BDG in broader generation settings such as summarization or code synthesis are promising directions for future applications of our work. That said:
>
> - We already include arithmetic free-form QA in GSM8K (Table 4), where BDG improves over both ECG and SCD despite being applied in a zero-shot setup (§3.3, L251–L256). GSM8K outputs are not multiple choice, and BDG operates over sampled spans from the LLM distribution P(y|x).
> - Our method supports open-ended candidate generation via nucleus or top-k sampling (L198), as noted in §3.1. These candidate sets are then re-ranked via game-theoretic interaction, and are not constrained to classification-style outputs.
> - Our Proposition 1 (L142–L153) extends to continuous confidence estimates, making BDG compatible with scoring in generative domains, provided candidate answer sets are well-defined (e.g., sampled summaries, code snippets, or rationales).
>
> We will clarify this generality in §3.1 and in the final appendix. We emphasize that BDG does not assume a task format: **it operates on any candidate set** where correctness likelihood can be estimated by a verifier.
>
> **3. Model recency**
>
> We will include results on newer models (e.g., LLaMA-3 and Qwen2.5) in the final version. The current evaluations (LLaMA-13B, DeepSeek-7B) were selected for reproducibility and baseline availability. BDG remains model-agnostic and does not depend on architecture-specific features or training.
>
> **4. Clarification on L198 — is *y* a token or sequence?**
>
> *y* refers to a candidate **sequence**, i.e., a full output sampled from the LLM. This is clarified in L198: “we construct *Y* by sampling candidates from the LLM’s distribution P<sub>LLM</sub>(y | x, correct).” These y ∈ Y are complete answers (e.g., strings), not individual tokens.
>
>
>
> ---
>
> We appreciate the reviewer’s thoughtful concerns and will strengthen our discussion of inference cost, open-ended generalization, and future directions. BDG is designed to balance correctness and consistency via strategic interaction at inference time, without introducing high sampling cost or retraining. We believe this makes BDG a practical and extensible framework for both structured and open-ended LLM decoding.

---

> > ### Comment · Reviewer_RFvm · 2025-08-05
> > **Official Comment by Reviewer**
> >
> > Thank you for your detailed response. Most of my concerns have been adequately addressed. I look forward to seeing results with newer models in the final version, and I will maintain my positive score.

---

### Official Review · Reviewer_Yhw4 · 2025-07-03

**Clarity:** 2
**Significance:** 3
**Originality:** 3
**Rating:** 4
**Confidence:** 3

**Summary:**

This paper proposes a game-theoretic decoding framework, called the Bayesian Decoding Game (BDG), to improve the self-verification capabilities of large language models (LLMs). The authors model decoding as a multi-stage interaction between a generator and a verifier, formalized as a signaling game, and introduce the concept of a $\sigma$-separated equilibrium to prevent collusion and enforce both correctness and consistency. Unlike existing methods such as single-agent self-reflection and multi-agent debate, which either suffer from confirmation bias or collusive agreement, BDG dynamically refines decoding strategies through no-regret learning and Markovian updates, aligning generation and verification outputs. Experiments across multiple benchmarks (e.g., MMLU, ARC, GSM8K) show that BDG consistently improves both factual accuracy and consistency, even enabling smaller models like LLaMA-13B to outperform much larger models (e.g., PaLM-540B), without requiring additional training. The method also shows robustness across tasks involving reasoning, ethics, and medical question answering.

**Questions:**

1. Is the verifier and generator the same LLM? Have you tried using different LLMs for the verifier and generator?

**Ethical Concerns:**

["NO or VERY MINOR ethics concerns only"]

**Final Justification:**

I hope the author can make the changes as promised in further version. I keep my positive score.

**Limitations:**

Yes

**Quality:**

3

**Strengths And Weaknesses:**

**Strengths**:
1. The paper introduces a new decoding strategy by framing LLM output generation as a multi-stage Bayesian signaling game between a generator and a verifier.
2. BDG operates at inference time and does not require fine-tuning or retraining. This makes it highly practical and easy to integrate into existing LLM pipelines.
3. The experimental results show that BDG yields consistent improvements across a variety of domains (e.g., ARC, MMLU, PubMedQA, Ethics) and question types (multiple choice and generative)

**Weaknesses**:
1. Although the method is theoretically sound, the practical convergence of the Markovian updates involves multiple hyperparameters (learning rate, stiffness, $\sigma$-thresholds), and the paper does not deeply explore their sensitivity or robustness.
2. The proposed method requires access to token probability, but the most performant LLMs are typically black-box models often with limited or no access to logits. It can not generalize to these black-box models. Therefore, some work has explored prompting language models to express uncertainty or confidence score in human language [2,3]. I think this paper should discuss and compare with this kind of method.
3. For the open-ended generative tasks, the authors should compare their method with various Best-of-N selection methods (such as self-consistency [1], universal self-consistency [2], soft self-consistency [3]) to show the effectiveness and efficiency of their method.
4. The method should be evaluated on more recent LLMs such as llama3 and qwen2.5 to show the generalization of their method.

[1] Self-Consistency Improves Chain of Thought Reasoning in Language Models (Wang et al., ICLR 2023)
[2] Universal Self-Consistency for Large Language Models (Chen et al., ArXiv 2023)
[3] Soft Self-Consistency Improves Language Models Agents (Wang et al., ACL 2024)

---

> ### Author Rebuttal · Authors · 2025-07-28
>
> **We thank Reviewer Yhw4 for their recognition of the theoretical soundness of our BDG formulation, its practicality as a plug-in decoding mechanism, and its consistent improvements across factual and reasoning tasks. Below we address each point raised.**
>
> ---
>
> **1. Hyperparameter robustness and rationale for fixed settings**
>
> We agree that further discussion of hyperparameter sensitivity would be valuable. Our primary focus in this work is the *framework-level design* of BDG as a decoding-time strategic mechanism that generalizes across models and domains. From this perspective, we intentionally avoid extensive task-specific tuning: excessive optimization of hyperparameters such as η, λ, or σ may improve local performance but reduce the generality and interpretability of the framework.
>
> In our experiments, we use a single fixed setting across all benchmarks and models (see Appendix I), and observe consistent convergence within 50–100 iterations (Fig. 3a). As the Markovian updates (Eq. 5–6) are entropy-regularized, convergence speed may vary with η, but the equilibrium ranking and preference alignment (Definition 3) remain stable. To support this point quantitatively, we will include an ablation over η and λ in the final appendix.
>
> **2. Applicability to black-box models and relation to prompt-based methods**
>
> We acknowledge that BDG requires access to token probabilities, which limits direct applicability to fully black-box APIs. However, we note that many black-box systems today are evaluated primarily via prompt engineering and meta-prompting techniques, which is an area orthogonal to our focus. BDG operates at the decoding layer and introduces an explicit multi-agent dynamic via policy updates; it does not rely on prompt design, instruction tuning, or post-hoc voting.
>
> While our goal is not to replace prompt-space heuristics, BDG can be adapted to use LLM-expressed confidence proxies or relative ranking signals (e.g., [2,3]) as substitutes for logits. We will discuss this adaptation path more explicitly in the limitations section.
>
> Specifically, in the updated Limitations section and Appendix we will include the following:
>
> - A clarification that BDG’s core mechanism only requires *relative preference rankings* or *confidence differentials* over candidates from both agents (generator and verifier).
> - A note that many LLM APIs now allow structured self-evaluation or scoring (e.g., “On a scale from 1 to 5, how confident are you in this answer?”). These can be used to construct soft $\sigma_i$-separation and entropy-regularized utility functions analogous to our token-based setting.
> - We will describe how BDG's Markovian belief updates (Eq. 5–6) can operate over *normalized confidence scores* instead of log-probabilities, which does not change the update logic.
> - We will include a pseudocode variant in Appendix J showing how the BDG belief update loop generalizes to black-box setups with prompt-accessible confidence outputs.
>
> These changes demonstrate that BDG can be *trivially extended* to black-box LLMs using elicited confidence without modifying the core dynamics of our framework.
>
> **3. Comparison with Best-of-N and self-consistency variants**
>
> We do not benchmark against Best-of-N variants (e.g., Self-Consistency [1], Universal Self-Consistency [2], Soft Self-Consistency [3]) because they follow a different design paradigm. These methods require large-scale sampling and rely on majority voting or heuristic filtering. In contrast:
>
> - BDG directly modifies the decoding procedure via a two-agent signaling game with provable convergence (Theorem 2, L128–L129), independent of CoT usage. Best-of-N variants cannot provide this level of independence and verifiability.
> - Our framework deterministically reweights a fixed candidate set through strategic interaction, achieving both consistency and correctness via equilibrium dynamics. We do not rely on post hoc aggregation as other methods do.
>
> We acknowledge these methods in the Related Work section and will clarify how BDG can complement them, e.g., by using BDG to select among CoT samples or self-consistent rationales (Table 4, §3.3).
>
> **4. Evaluation on newer models**
>
> Our main experiments use open-weight models (LLaMA-7B/13B and DeepSeek-7B) to allow fair comparison against baseline decoders such as ECG [31], MI, and SCD. However, we emphasize that BDG is *model-agnostic*: it applies to any autoregressive LLM with scoring access and does not require retraining or internal architectural modification.
>
> We demonstrate this with DeepSeek-7B on MMLU, where BDG improves performance from 44.07% (base) to 49.52%, surpassing ECG at 46.20% (§3.2, L244–L246). We are currently seeing similar results for BDG with LLaMA-3 and Qwen2.5 models, and these results will be included in the final appendix.
>
> **5. Mixing of Heterogeneous Models**
>
> We did not mix different models by default, because doing so introduces asymmetry that dilutes the game‑theoretic meaning of the interaction, where differences in model capability and calibration can dominate outcomes rather than the strategic dynamics we study, thus changing the problem class to a more LLM-as-a-judge/student-teacher like topic. Thus, exploring controlled model asymmetry (e.g., pairing a specialized verifier with a domain‑specific generator under explicit calibration checks) is a valuable direction to investigate it as future work, and we thank Reviewer Yhw4 for the thoughtful suggestion.
>
> ---
>
> We appreciate the reviewer’s constructive feedback. BDG provides a decoding-time, model-agnostic mechanism that formalizes generator-verifier dynamics via Bayesian signaling games, offering provable convergence to a separating equilibrium. We will incorporate the suggested clarifications on robustness, black-box adaptation, and comparison with sampling-based approaches in the final version.
>
> ---
>
> **References**
> [1] Wang et al., "Self-Consistency Improves Chain of Thought Reasoning in Language Models", ICLR 2023. arXiv: 2203.11171
>
> [2] Chen et al., "Universal Self-Consistency for Large Language Models", arXiv 2023. arXiv: 2311.17311
>
> [3] Wang et al., "Soft Self-Consistency Improves Language Model Agents", ACL 2024. arXiv: 2402.13212

---

> > ### Comment · Reviewer_Yhw4 · 2025-08-08
> >
> > Thanks for the detailed response. Most of my concerns are solved. I hope the author can make the changes as promised in further version. I keep my positive score.

---

> ### Comment · Area_Chair_SqsA · 2025-08-07
> **Please respond to authors.**
>
> Dear Reviewer,
>
> Your active participation in the review process is crucial. Please respond to the authors' response and acknowledge you have done so.
>
> Thanks.
>
> -AC

---

### Official Review · Reviewer_YiqR · 2025-07-03

**Clarity:** 2
**Significance:** 1
**Originality:** 3
**Rating:** 4
**Confidence:** 3

**Summary:**

This paper aims to address the issue of consistency or flip-flopping and correctness when large language models reason. The paper outlines the trade-off between inconsistency in single agent setting as well as a Nash equilibrium collapse in multi-agent setting where models converge to the incorrect answer. Therefore, in order to build consensus during inference, this paper proposes formulating this as a Bayesian Decoding Game and proposes an optimization algorithm to achieve this. The paper tests this hypothesis on a variety of natural language tasks and (older) language models.

**Questions:**

See weaknesses above.

**Ethical Concerns:**

["NO or VERY MINOR ethics concerns only"]

**Final Justification:**

I think the authors have mentioned to include the additional works in their related works section and sharpen their contributions and contrast. I also think in the camera-ready they should experiment with more recent models, but I think the paper can be accepted.

**Limitations:**

Yes

**Quality:**

3

**Strengths And Weaknesses:**

**Strengths**
- Addresses an important connection between consistency and correctness as two desirable properties of good reasoning systems.
- The decoding method is fairly novel and outperforms the baselines

**Weaknesses**
- Missing citation to related work: Several works have addressed the issue of inconsistency in language models which the paper does not address or cite:
    - https://arxiv.org/abs/2203.11171 (this paper also uses consensus to improve reasoning performance, so why not compare against it?)
    - https://arxiv.org/abs/2311.08596
    - https://arxiv.org/abs/2212.09251
- Missing explanation of baselines: The experimental section of the paper does not outline the choice of baseline decoding methods. What do MI, D, SCD methods stand for? Why not use few-shot evaluation for reasoning tasks like GSM-8K where the reported numbers are exceptionally low and it is well-known that chain-of-thought sampling yields high improvement boosts. Why only include zero-shot and few-shot evaluation in Table 3 and why not report other decoding baselines apart from BDG.
- Missing justification for choice and models. Clearly the models used in this paper are a few generations behind and obsolete, making some of the results not highly convincing. The paper fails to address this and show gains from their decoding method still persist with recent generation of models and with chain of thought sampling.

---

> ### Author Rebuttal · Authors · 2025-07-28
>
> **We thank Reviewer YiqR for highlighting the connection between consistency and correctness, and for noting that our decoding method is novel and outperforms strong baselines. We address the reviewer’s concerns below by clarifying the positioning of our work and its distinction from the cited literature, and by reaffirming our key contributions.**
>
> ---
>
> **1. On the relation to cited works**
>
> The reviewer recommends comparing against three recent works [1,2,3]. While we appreciate the relevance of these papers to the general topic of reasoning and consistency, they differ substantially from our proposal in both goal and method.
>
> Rather than sampling diverse rationales or probing sycophancy in multi-turn dialogue, we study *decoding-time strategic consensus* with formal game-theoretic guarantees. Concretely:
>
> - The **first paper** proposes a prompt + sampling strategy (sample-and-marginalize over CoT outputs). In contrast, our BDG modifies decoding itself via a two-player Bayesian game between a generator and verifier that provably converges to a *σ-separated equilibrium* (Definition 3, L107–L111). This equilibrium enforces strategic separation to prevent collusion and aligns verifier/generator preferences via Markovian updates (Eq. 5–6).
>
> - **BDG is formal, training-free, and model-agnostic.** It does not rely on specific prompting formats or auxiliary supervision, and can be composed with CoT/few-shot prompting orthogonally (see Table 3–4 and Appendix I). Our contribution goes beyond deliberation or contrastive-objective methods.
>
> - The **second and third papers** primarily *diagnose* LLM behaviors (e.g., inverse scaling, multi-turn sycophancy), but do not propose a decoding-time algorithm that enforces reliable output or prevents collusion. They offer no equilibrium or strategy formulation, nor convergence guarantees. Their contribution lies in behavioral analysis, not decoding design.
>
> We will discuss these differences in detail in the Related Works section and will cite the provided references.
>
> **2. Clarification of baselines and experimental scope**
>
> We define all baseline decoders in Appendix I (§I), including:
> **G** (Generative ranking), **D** (Verifier-only ranking), **MI** (Mutual Information reweighting), **SCD** (Self-Contrastive Decoding), **ECG** (Equilibrium Consensus Game), and our proposed **BDG**.
>
> All methods operate over the same candidate set, models, and decoding budget, ensuring fair comparison. BDG shows consistent improvements across MMLU, ARC-E/C, RACE-H, GSM8K, and TruthfulQA, often allowing LLaMA-13B to match or exceed much larger models like PaLM-540B (see Table 1, Table 2, §3.2).
>
> **3. On model choices and extensibility**
>
> BDG is explicitly model-agnostic. It only requires access to model likelihoods for reweighting and converges within a small number of steps (typically <100; see Fig. 3a). While we use LLaMA-7B/13B and DeepSeek-7B for comparison with baselines like ECG [31], our framework applies to newer models as well.
>
> In fact, BDG improves DeepSeek-7B on MMLU from 44.07% to 49.52%, outperforming both its base and ECG variants (46.20%) — see §3.2 (L244–L246). Additional experiments on newer models will be included in the final version.
>
> **4. Our Core Contributions (beyond prior art)**
>
> - **Theoretical novelty**: We are the first to model LLM decoding as a *Bayesian signaling game* with a provably convergent equilibrium (Theorem 2, L128–L129), addressing multi-agent collusion and self-confirmation failures (Appendix C, Fig. 3).
>
> - **Decoding-time optimization**: Our Markovian update schedule defines no-regret learning between generator and verifier, ensuring preference alignment and stability (Fig. 4a).
>
> - **Training-free and plug-and-play**: BDG improves decoding without additional data, training, or architectural modification. Its consensus enforcement is robust to prompt format, model size, and domain (Tables 1–5).
>
> - **Separation over heuristics**: Unlike prior sampling or self-assessment heuristics, BDG’s correctness–consistency tradeoff is controlled analytically via *σ-separation* and preference alignment metrics (Definition 3; Proposition 1).
>
> ---
>
> We respectfully submit that the cited works operate under different assumptions, problem formulations, and technical mechanisms. BDG is a decoding-time game-theoretic framework with theoretical guarantees and consistent empirical benefits. We will clarify these distinctions more explicitly in the Related Work section and appreciate the reviewer’s feedback in helping us sharpen the description of our contributions.
>
> ---
>
> **References**
>
> [1] Wang et al., "Self-Consistency Improves Chain of Thought Reasoning in Language Models", ICLR 2023. arXiv: 2203.11171
> [2] Laben etal., "Are You Sure? Challenging LLMs Leads to Performance Drops in The FlipFlop Experiment", arXiv: 2311.08596
> [3] Perez et al."Discovering Language Model Behaviors with Model-Written Evaluations
> ", ACL Findings 2023. arXiv: 2212.09251

---

> > ### Comment · Reviewer_YiqR · 2025-08-03
> >
> > Thanks, I am convinced with the arguments made by the author. With updates to related work and perhaps more experimentation with recent crop of models in the camera-ready, I would be happy to see this paper accepted.

---

### Note · Authors · 2025-08-14

Dear reviewers and ACs,

We sincerely thank all reviewers for their careful evaluations and constructive feedback. We appreciate Reviewer YiqR's recognition of the novelty of our decoding method and its empirical improvements, as well as the thoughtful suggestions on related work and evaluation scope. We thank Reviewer Yhw4 for acknowledging the theoretical soundness and practicality of BDG, and for raising valuable points on robustness, applicability to black-box models, and comparisons with sampling-based approaches. We are also grateful to Reviewer RFvm for highlighting the clarity of our framework and its substantial gains, and to Reviewer sYwu for recognizing the theoretical contribution, equilibrium formulation, and consistent empirical performance.

In our rebuttal, we clarified the distinctions from related methods, expanded the definition and fairness of baselines, and demonstrated that BDG is model-agnostic and applicable across both structured and generative tasks. We also explained its efficiency, convergence behavior, and extensibility to newer models and black-box settings, and outlined additional experiments and ablations to be included in the final version. Across all major points, our responses show that BDG offers a theoretically grounded, training-free, and practically deployable decoding-time mechanism that provably balances correctness and consistency through game-theoretic interaction.

The reviewers' comments have strengthened our presentation and reinforced our belief that this work addresses a critical gap in LLM decoding. By combining formal guarantees with consistent empirical gains, it advances both the theoretical and practical understanding of strategic consensus in language models. We look forward to refining the manuscript accordingly and trust the revisions will make the contribution clearer and more complete for the final version.

---

### Decision · Program_Chairs · 2025-09-17

**Decision:**

Accept (poster)

**Comment:**

The paper introduced a new decoding method, which they refer to as a multistage Bayesian Decoding Game. The aim of the game is to decrease logical inconsistency and increase the change of arriving at the correct answer when using multi-agent inference to solve some task. Their method involves LLM-based generators and verifiers that refine their strategies over multiple iterations of the game.

The proposed method merges desires for consistency and correctness in the answers from AI systems. Empirically, it seems to work very well on the tasks that were evaluated. All reviewers expressed support for the value of the proposed method, especially since it is inference-time only and doesn't require any additional model training.

Reviewers did, however, note several weaknesses. Multiple reviewers expressed concerns about the baseline methods. Existing baselines were insufficiently documented in the main paper body. Also, more baselines (for example, a best-of-n) should have been considered. Evaluation was only conducted on tasks with simple numerical or classification-style answers, so we don't know what performance will be like on harder tasks such as summarization or code generation.

Overall, the paper offers a reasonably-sized contribution which I expect will be of interest to NeurIPS community. I strongly advise the authors to make the various textual changes recommended by reviewers (expanded references to related work, clearer description of baselines in main paper, more info on computational cost of inference of proposed method vs. other methods, improved accessibility for readers unfamiliar with game theory).